



# Reanalysis comparisons of upper tropospheric/lower stratospheric jets and multiple tropopauses

Gloria L. Manney[1,2], Michaela I. Hegglin[3], Zachary D. Lawrence[2], Krzysztof Wargan[4], Luis F. Millán[5], Michael J. Schwartz[5], Michelle L. Santee[5], Alyn Lambert[5], Steven Pawson[4], Brian W. Knosp[5], Ryan A. Fuller[5], and William H. Daffer[5]

[1]NorthWest Research Associates, Socorro, NM USA
[2]New Mexico Institute of Mining and Technology, Socorro, NM USA
[3]University of Reading, Reading, United Kingdom
[4]NASA/Goddard Space Flight Center, Greenbelt, MD USA
[5]Jet Propulsion Laboratory, California Institute of Technology, Pasadena, CA, USA

*Correspondence to:* Gloria L Manney (manney@nwra.com)





**Abstract.** The representation of upper tropospheric/lower stratospheric (UTLS) jet and tropopause characteristics is compared in five modern high-resolution reanalyses for 1980 through 2014. Climatologies of upper tropospheric jet, subvortex jet (the lowermost part of the stratospheric vortex), and multiple tropopause frequency distributions in MERRA (Modern Era Retrospective Analysis for Research and Applications), ERA-I (the ECMWF interim reanalysis), JRA-55 (the Japanese 55-year Reanalysis), and CFSR (the Climate Forecast System Reanalysis) are compared with those in MERRA-2. Differences between alternate products from individual reanalysis systems are assessed; in particular, a comparison of CFSR data on model and pressure levels highlights the importance of vertical grid spacing. Most of the differences in distributions of UTLS jets and multiple tropopauses are consistent with the differences in assimilation model grids and resolution: For example, ERA-I (with coarsest native horizontal resolution) typically shows a significant low bias in upper tropospheric jets with respect to MERRA-2, and JRA-55 a more modest one, while CFSR (with finest native horizontal resolution) shows a high bias with respect to MERRA-2 in both upper tropospheric jets and multiple tropopauses. Vertical temperature structure and grid spacing are especially important for multiple tropopause characterization. Substantial differences between MERRA and MERRA-2 are seen in mid- to high-latitude southern hemisphere winter upper tropospheric jets and multiple tropopauses, and in the upper tropospheric jets associated with tropical circulations during the solstice seasons; some of the largest differences from the other reanalyses are seen in the same times and places. Very good qualitative agreement among the reanalyses is seen between the large scale climatological features in UTLS jet and multiple tropopause distributions. Quantitative differences may, however, have important consequences for transport and variability studies. Our results highlight the importance of considering reanalyses differences in UTLS studies, especially in relation to resolution and model grids; this is particularly critical when using high-resolution reanalyses as an observational reference for evaluating global chemistry climate models.

# 1 Introduction

Variations in the upper tropospheric/lower stratospheric (UTLS) jets and extratropical tropopause influence high-impact weather and climate on regional and global scales: They play key roles in circulation changes, especially the observed widening of the tropics (e.g., Staten et al., 2016) and storm track evolution (Barnes and Screen, 2015; Messori et al., 2016; Woollings et al., 2016, and references therein). They influence surface weather patterns (e.g., see reviews by Lucas et al., 2014; Harnik et al., 2016) such as rainfall changes (e.g., Price et al., 1998; Raible et al., 2004; Karnauskas and Ummenhofer, 2014; Huang et al., 2015; Xie et al., 2015; Delworth and Zeng, 2014; Bai et al., 2016), destructive wind storms (e.g., Pinto et al., 2009, 2014; Gómara et al., 2014; Messori and Caballero, 2015), and extreme temperature events (e.g., Francis and Vavrus, 2012; Harnik et al., 2016). Moreover, transport processes that alter the extent and consequences of extra-tropical stratosphere-troposphere exchange (STE) are closely linked to the tropopause and jets, which are themselves sensitive to climate change and ozone depletion (e.g., Seidel and Randel, 2006; Lorenz and DeWeaver, 2007; Polvani et al., 2011; WMO, 2011; Hudson, 2012; Grise et al., 2013; Waugh et al., 2015). Both tropospheric and total column ozone vary with tropopause height and STE near the UTLS jets (e.g., Olsen et al., 2002; Neu et al., 2014), as well with natural modes of variability such as ENSO that alter the jets





(Hudson, 2012; Lin et al., 2014, 2015; Olsen et al., 2016, and references therein). Thus, much of the variability in UTLS ozone is inextricably linked to that of the UTLS jets.

Modern high-resolution reanalyses from data assimilation systems produced by European Centre for Medium-range Weather Forecasts (ECMWF), the National Aeronautics and Space Administration's (NASA's) Global Modeling and Assimilation Of-
fice (GMAO), the National Centers for Environmental Prediction (NCEP), and the Japanese Meteorological Agency (JMA) are invaluable tools for studying and understanding UTLS dynamical and transport processes. It is only the latest generations of these reanalyses that provide products on the full model grids that can resolve many of the regionally and rapidly varying dynamical processes in the UTLS. While high-resolution datasets such as those from sondes and Global Positioning System-Radio Occultation provide critical insights on the structure of the extra-tropical tropopause region, no available data sources
can provide the global time-resolved fields, including winds, that reanalyses provide that are necessary to understand the global effects of jet and tropopause variations. Reanalyses are thus a critical tool for UTLS studies, and are also widely used as an observational reference for climate model intercomparisons (e.g., Gettelman et al., 2010). However, they are also highly dependent on the details of the underlying general circulation models and assimilation systems, as well as on the input datasets and processing. Several previous studies have shown differences in upper tropospheric jet and/or tropopause information from
multiple reanalyses (e.g., Archer and Caldeira, 2008; Pena-Ortiz et al., 2013; Boothe and Homeyer, 2016). Studies of tropical width using metrics related to zonal mean upper tropospheric jets and/or the tropopause have shown inconsistent results between models and reanalyses, as well as among reanalyses (e.g., Davis and Rosenlof, 2012; Davis and Birner, 2017). Most of these studies have used older reanalyses or focused on tropopause and/or jet diagnostics based on zonal means. Pena-Ortiz et al. (2013) used a three-dimensional (3D) jet characterization scheme, but applied it to the NCEP/NCAR Reanalysis and the
NCEP "20th Century" reanalysis (the latter assimilates only surface observations), both of which use relatively unsophisticated or outdated assimilation systems, have coarse horizontal resolution and poor vertical resolution in the UTLS, and have been shown to have limited skill in the UTLS and above (see Fujiwara et al., 2017, for a review of reanalysis system characteristics and evaluations). While Davis and Birner (2017) used four of the five modern reanalyses we will compare here, their tropopause and jet-based tropical width diagnostics were based on analysis of zonal mean fields.

Manney et al. (2011) developed a method for characterizing the upper tropospheric jets, the stratospheric subvortex jet, and multiple tropopauses. Manney et al. (2014) used this package to present a detailed climatology of these UTLS jets and multiple tropopauses, and the relationships between them, using GMAO's Modern Era Retrospective-analysis for Research and Applications (MERRA). Here we evaluate the representation of these climatological features in the four other most recent high-resolution reanalyses: MERRA-2 (the successor to MERRA), ECMWF's ERA-Interim, JMA's JRA-55, and NCEP's Cli-
mate Forecast System Reanalysis (CFSR); comparisons of MERRA-2 with its predecessor are also included. These diagnostics cannot be directly compared with observations, and thus reanalysis comparisons are a unique tool to help assess the robustness of and uncertainties in the representation of UTLS dynamical features in reanalyses. Section 2 describes the reanalysis datasets and the methods used. In Section 3.1 we evaluate differences between several commonly used configurations of and output products from several of the reanalyses. Section 3.2 provides a comparison of seasonal upper tropospheric jet, multiple





tropopause, and subvortex jet distributions, while Section 3.3 compares the climatological annual cycles of these fields among the reanalyses. A summary and conclusions are presented in Section 4.

## 2 Data and Methods

### 2.1 Reanalysis Data

5   The reanalysis datasets used here are briefly described below. Detailed descriptions of the models, assimilation systems, and data inputs for each are given in the overview paper on the Stratosphere-troposphere Processes And their Role in Climate-Reanalysis Intercomparison Project (S-RIP) (Fujiwara et al., 2017). The five recent high-resolution "full-input" reanalysis climatologies are compared for 1980 through 2014, with the December-January-February, DJF, seasonal plots starting with December 1979. All analyses are done using daily 12-UT fields from each reanalysis dataset.

### 10   2.1.1   MERRA and MERRA-2

The National Aeronautics and Space Administration (NASA) GMAO's MERRA (Rienecker et al., 2011) dataset is a global reanalysis covering 1979 through 2015. It is based on the GEOS (Goddard Earth Observing System) version 5.2.0 assimilation system, which uses 3D-Var assimilation with Incremental Analysis Update (IAU) (Bloom et al., 1996) to constrain the analyses. The model uses a $0.5° \times 0.667°$ latitude/longitude grid with 72 hybrid sigma-pressure levels, with about 0.8 km vertical spacing

in the upper troposphere, increasing to $\sim$1.2 km in the UTLS. The fields used here are provided on the model grid.

  MERRA-2 (Gelaro et al., 2017) uses a similar model and assimilation system to MERRA, with updates also described by Bosilovich et al. (2015), Molod et al. (2015), and Takacs et al. (2016); the data products are described by Bosilovich et al. (2016). All MERRA-2 data products used here are on model levels (the same vertical grid as for MERRA) and a $0.5° \times 0.625°$ latitude/longitude grid. Data from MERRA-2 from its spin-up year, 1979, are not in the public MERRA-2 record; we do,

however, use December data from that year to construct the DJF climatologies.

  For MERRA-2, GMAO provides "Analyzed" (ANA) and "Assimilated" (ASM) file collections (Global Modeling and Assimilation Office (GMAO), 2015b, a, respectively). As described by Fujiwara et al. (2017), the ANA fields are written after the analysis step, but before the IAU is applied; these products are analogous to the analyzed fields produced by other reanalysis centers (e.g., Fujiwara et al., 2017). The ASM output is the product of IAU written by the general circulation model forced by

the analysis increments computed in the analysis step. The GMAO recommends the ASM file collection for most purposes, because it provides the most dynamically consistent set of fields, as well as a fuller set of atmospheric variables. For MERRA, however, the ASM fields are not available on the model grid, but only at degraded horizontal and vertical resolution; because of the importance of resolution to UTLS studies, we thus use the MERRA ANA collection here. Differences between ANA and ASM fields are small, but can be non-negligible (see Section 3.1).



### 2.1.2 ERA-Interim

ERA-Interim (see Dee et al., 2011) is another global reanalysis that covers the period from 1979 to the present. The data are produced using 4D-Var assimilation with a $T_L 255 L 60$ spectral model. Here we use the data on a $0.75° \times 0.75°$ latitude/longitude grid (near the resolution of the model's Gaussian grid) on the 60 model levels. The spacing of the model levels in the lower stratosphere is $\sim 1$ km.

### 2.1.3 JRA-55

JRA-55 (Ebita et al., 2011; Kobayashi et al., 2015) is a global reanalysis that covers the period from 1958 to the present. The data are produced using 4D-Var assimilation with a $T_L 319 L 60$ spectral model. We use the fields on the model grid and vertical levels, which has a resolution of $\sim 1$ km in the UTLS. A reanalysis, JRA-55C, using the same assimilation system as for JRA-55, but with only "conventional" data inputs (that is, no satellite data) was run for 1972 through 2012 (Kobayashi et al., 2014; Fujiwara et al., 2017). In Section 3.1 we compare results for JRA-55 and JRA-55C for 1979 through 2012 (during the "satellite era").

### 2.1.4 CFSR

NCEP-CFSR/CFSv2 (hereinafter CFSR) (Saha et al., 2010) is a global reanalysis covering the period from 1979 to the present. The data are produced using a coupled ocean-atmosphere model and 3D-Var assimilation. The model resolution is T382L64; the data used here are on a $0.5° \times 0.5°$ horizontal grid on the model levels (available through 2014); vertical resolution in the UTLS is near 1 km. These model level data have only recently been made available; prior to that, the NCEP $0.5° \times 0.5°$ data were provided only on a vertical grid with 37 pressure levels between 1000 and 1 hPa, resulting in a vertical spacing near 2 km in the UTLS; in Section 3.1 we compare pressure and model level fields to illustrate the importance of vertical resolution.

### 2.2 Jet and Tropopause Characterization

The methods used here to characterize the jets and tropopauses are those of Manney et al. (2011, 2014). At each longitude, an upper tropospheric jet core is identified at every latitude and vertical grid-point where the windspeed maximum exceeds 40 m/s. The boundaries of the jet region are the four grid-points vertically above and below and horizontally poleward and equatorward of the core where the windspeed drops below 30 m/s. When more than one windspeed maximum greater than 40 m/s appears within a given 30 m/s contour, they are defined as separate cores if the latitude distance between them is greater than 10° or the value of the minimum windspeed on the line between them is at least 30 m/s less than the windspeed value at the strongest core. These parameters were tuned to approximate as closely as feasible the choices that would be made by visual inspection.

The subvortex jet core is identified as the most poleward maximum in westerly windspeed at each model level that exceeds 30 m/s, and the locations of the 30 m/s contour crossings poleward and equatorward of this define the boundaries of the subvortex jet region. The bottom of the subvortex jet often extends down to the top levels of the upper tropospheric jets. To distinguish





between the two in such cases, we first identify the subvortex jet at levels down to a pressure near 300 hPa. We then work down from the model level nearest 80 hPa to identify the lowest altitude at which the windspeed of the jet is still decreasing with decreasing altitude; this is defined as the bottom of the subvortex jet. "Merged" subvortex and upper troposphere jets are identified as those where the bottom of the subvortex jet region is not separated from the top of an upper tropospheric jet region.

Maps of subvortex jet frequency distributions use the latitude at the minimum altitude as the position of each subvortex jet identified.

The thermal (temperature gradient) tropopause is calculated using the World Meteorological Organization (WMO) definition, wherein dT/dz must rise above $-2$ K/km and remain about that on average for at least 2 km (see, e.g., Homeyer et al., 2010, for review and discussion of issues related to calculating the thermal tropopause). Multiple thermal tropopauses are

identified if dT/dz drops below $-2$ K/km above the primary thermal tropopause, as the next level where the WMO criteria are again fulfilled (e.g., Randel et al., 2007; Manney et al., 2011, 2014).

## 2.3   Comparison methodology

The bulk of the comparisons presented here are of frequency distributions, calculated as described in more detail by Manney et al. (2014). Because it doesn't make sense to construct means of frequency distributions from multiple reanalyses, we have

chosen to compare all other reanalyses to MERRA-2. MERRA-2 was chosen because it is the most recent of the modern high-resolution reanalyses and thus the comparisons extend the evaluation of this new reanalysis dataset. We show frequency distributions from MERRA-2, and differences between those distributions and MERRA-2 for the other reanalyses. Because the frequency distributions are expressed as a percent, the absolute differences (i.e., $\text{Freq}_{r1} - \text{Freq}_{r2}$, where $r1$ and $r2$ are two reanalyses) between two frequency distributions that are shown in the figures are also in units of percent; this should not be

confused with the approximate percentage values for relative differences (e.g., $(\text{Freq}_{r1} - \text{Freq}_{r2})/(\text{Freq}_{r1} + \text{Freq}_{r2}) \times 100$) mentioned in the text.

To directly compare frequency distributions from reanalyses on different grids, we construct the 2D histograms using the same bins for each reanalysis. Comparing frequency distributions for "threshold" phenomena such as the existence of jets or multiple tropopauses is problematic. In general, we characterize the jets and tropopauses on the high-resolution lati-

tude/longitude grids of the reanalysis datasets. These characterizations are then used to calculate 2D histograms within wider latitude/longitude bins. In the following description "gridpoints" refers to the reanalysis grid, and "bins" to the coarser latitude/longitude grid on which the 2D histograms are constructed. The issue of consistent normalization is relatively straightforward: The normalization procedure used herein is similar to that described by Manney et al. (2014), but for each reanalysis, we calculate the number of gridpoints that would "fill" each individual bin based on the bin size and the reanalysis grid spacing;

the total counts in each bin are then divided by this value. The upper tropospheric and subvortex jet distributions are normalized by the total number of longitude gridpoints in each bin since the definition of the jets makes it extremely unlikely that more than one jet at the same longitude would be in the same bin: For example, upper tropospheric jets must be separated by either a drop in windspeed to below 30 m/s or 10° in latitude; with latitude bin size of 3 or 4° (the values used here and in Manney et al., 2014, respectively), to have two jets at one longitude in a single bin would require exceptionally strong windspeed gra-





dients in a region where the jet core windspeed was just above the 40 m/s threshold. The multiple tropopause distributions are normalized by the total number of gridpoints (latitudes by longitudes) that are in each bin since the profile at each gridpoint has the potential to have more than one tropopause.

Beyond this, however, aliasing discrepancies arise in cases where a strong localized (particularly in latitude) feature lies near the boundary of a bin. In such cases, the differences between the reanalysis grid point locations with respect to the bin edges can result in counts (such as existence of a jet core or multiple tropopause) falling preferentially in one bin in one reanalysis and in the adjacent bin in another reanalysis. This problem is not substantially improved for jet distributions (identified in part by the latitude of the maxima) by interpolating to a common latitude grid, because that interpolation can lead to similar problems wherein the maximum of the interpolated field can be preferentially shifted in one direction depending on the relative spacing of the interpolated and uninterpolated grids. We have found that choosing a latitude bin size such that an integer number of reanalysis gridpoints fits into the bin practically eliminates this difficulty. For JRA-55 and JRA-55C, where the data are provided on an approximately $0.5625°$ Gaussian grid, we chose to interpolate to a $0.5°$ latitude grid before doing the jet and tropopause identification analysis. This grid is sufficiently close to the native grid that aliasing of a jet core (location of maximum in windspeed) by the interpolation is uncommon. Throughout this paper, we use $3°$ latitude and $6°$ longitude bins for maps, and $3°$ latitude and 1 km altitude bins for cross-sections. When our histograms constructed with the "matched" bin sizes are normalized by the maximum in the frequency distribution for each reanalysis (thus eliminating information on the difference in maximum frequency between reanalyses), the results show nearly identical patterns to those using the normalization described above, suggesting that our normalization procedure is robust.

For altitude/latitude cross-sections, because there is no obvious way to define the number of vertical gridpoints that "fill" a bin (because the relationship of model levels to bin locations varies with time and geographical location), we have chosen not to normalize by vertical spacing. The 1 km vertical bin size used here is chosen to include approximately one vertical grid point at each latitude/longitude. This is of little consequence for upper tropospheric jets and multiple tropopauses, where there is one vertical location identified for each feature. For the subvortex jets, which are identified at each level, we will show some differences that arise from the relationship between different model vertical grids and bin size. While one might argue that these are merely an artifact of the analysis procedure, they do provide information on the limitations of the information content of the reanalysis fields as provided to users.

## 3   Results

### 3.1   Grid, Output Product, and Assimilated Field Choices

Most reanalysis centers provide products on several different grids, in particular both on model levels and interpolated to a coarser set of standard pressure levels. In addition, they provide different types of output datasets and sometimes alternate reanalyses based on limited input datasets. We explore here the results of some of these choices of which product to use.

Products available from MERRA-2 include those from the ANA and ASM collections, as described above and by Bosilovich et al. (2016). While the ASM products are recommended by GMAO for most studies, this distinction has not been widely recog-





nized, so usage of one rather than the other has been inconsistent in existing studies. Furthermore, ASM products for MERRA were only available on a reduced-resolution grid – interpolated both to a coarser horizontal grid and pressure levels with coarser vertical spacing. Figures 1 and 2 show the differences between frequency distributions from MERRA-2 ASM and ANA for September-October-November (SON) for upper tropospheric jets, multiple tropopauses, and subvortex jets. SON was chosen

to illustrate characteristic differences seen in both hemispheres; differences are generally slightly larger in the winter solstice season in each hemisphere, and smaller (or undefined in the case of subvortex jets) in the summer solstice season. Differences are generally small (less than about 5% of the maximum MERRA-2 frequencies for upper tropospheric and subvortex jets, and about 10% of the maximum MERRA-2 frequencies for tropopause locations and multiple tropopause frequencies). Systematic differences include a slight northward shift of both northern hemisphere (NH) and southern hemisphere (SH) subtropical jets

(top row of Figure 1) and of the SH subvortex jet (bottom row of Figure 1) in ASM versus ANA fields. The NH subvortex jets show a pattern of alternating negative and positive differences near the pole, which is even more pronounced in DJF (not shown); this is a known artifact that arises because the horizontal wind vector in the ASM fields is remapped from the model's cubed-sphere grid to a latitude-longitude grid, whereas the ANA fields are produced by the analysis module, which uses a latitude-longitude grid (Bosilovich et al., 2015). The top row of Figure 2 indicates that the poleward shift of the NH subtrop-

ical jet (centered near 30°N) in ASM versus ANA is accompanied by a downward shift of about a kilometer; small negative differences near 40°N below this jet suggest this may partly be due to a narrowing of its vertical extent. There is a lower incidence of multiple tropopauses in ASM versus ANA (Figure 1, second row; Figure 2 third and fourth rows). The second row of Figure 2 (single tropopause locations) indicates a downward shift of the tropical tropopause in ASM versus ANA. While all of the ASM/ANA differences are small, they are often systematic. To the extent that the MERRA and MERRA-2 models and

assimilation systems are similar, these differences may help indicate the level of differences that might have been seen if ASM fields were available for MERRA.

The CFSR dataset, for which model-level fields have only recently been made available, is used to illustrate the importance of vertical grid spacing for jet and tropopause characterization. Figures 3 and 4 compare jet and tropopause frequency distributions between model and pressure level CFSR fields for SON on the same horizontal grids. The pressure level data show a small

but significant (up to about 10% of the maximum frequencies seen in the model level data) global decrease in the number of upper tropospheric jet cores detected (Figure 3, top row). Figure 4 (top row) shows an oscillatory pattern in the altitudes of the jets that are identified between the model level and pressure level data. The patterns in both figures suggest that jets are often mis-located in the vertical, and may be missed entirely where the spacing of the pressure levels is such that the maximum windspeed on those levels does not exceed the 40 m/s threshold. It is also unsurprising that a vertical spacing near 2 km in

the UTLS for the pressure level data results in many fewer multiple tropopause identifications, and consequently more single tropopause identifications (Figure 3, second row, and Figure 4, second through fourth rows). The multiple tropopauses, and mid- to high-latitude single tropopauses, that are identified in the pressure level data appear on average to be close to the same altitude as those in the model level data. The single tropical tropopause shows a low altitude bias. The pressure level results show a small deficit (seen as positive values) in the total number of subvortex jets (Figure 3, third row), with a dipole pattern

suggesting systematic shifts in the position; this shift likely arises because the stratospheric vortex typically increases in area,





and also tilts, with height, both of which change the latitude of the subvortex jet demarking its edge (these changes with height are especially pronounced in the disturbed conditions during NH fall and winter, consistent with the large NH differences over Asia and the western Pacific). The pressure level data show a marked surplus of merged subvortex and upper tropospheric jets (Figure 3, bottom row), because those are identified by comparing the vertical gradient in windspeed at adjacent levels, and the coarser resolution misses levels that are in neither jet region.

Several of the reanalysis centers have produced "conventional data only" (i.e., no satellite data inputs) reanalyses (for an overview, see Fujiwara et al., 2017). The JMA's JRA-55C is such a reanalysis for 1972 through 2012 using the same model and assimilation system as for JRA-55 (Kobayashi et al., 2014). To elucidate the impact of including satellite data in the assimilation during the period since 1979 that we study here (often referred to as the "satellite era"), Figures 5 and 6 show the JRA-55/JRA-55C differences for June-July-August (JJA) (again, the season is chosen to show the most characteristic behavior). The SH extratropical differences are much larger than those in the NH in all seasons, as expected given the dearth of conventional data in the SH; especially, NH subvortex differences are very small even in DJF (not shown). Both the subtropical and polar upper tropospheric jets (Figure 5, top row) show an equatorward shift in JRA-55C with respect to JRA-55, which is consistently seen in all seasons. The SH polar jet shows a consistent upward and poleward shift in JRA-55C with respect to JRA-55 (Figure 6, top row). The differences between JRA-55 and JRA-55C multiple tropopauses in JJA show a longitudinal dipole pattern poleward of 60°S, with more multiple tropopauses in JRA-55 than in JRA-55C in the western hemisphere and an opposite pattern with fewer multiple tropopauses in JRA-55 than in JRA-55C in the eastern hemisphere. In March-April-May (MAM) (not shown) this same pattern appears, but without the global band of higher multiple tropopause frequencies in JRA-55 near 40–60°S. Multiple tropopauses at high latitudes have higher secondary tropopauses (Figure 6, fourth row) in JRA-55C, and single tropopauses (Figure 6, second row) are lower in SH high latitudes. The SH subvortex jets are consistently shifted equatorward in JRA-55C with respect to those in JRA-55 during all seasons when they are present (Figure 5, third row).

The above results illustrate the consequences of some of the choices of products from a given reanalysis center. Some of these differences are large enough to have a significant impact on zonal mean quantities calculated from these datasets, with multiple tropopause characteristics being particularly sensitive to the reanalysis configuration. In the following sections, we evaluate in detail the differences in upper tropospheric/subvortex jet and multiple tropopause climatologies from the most recommended and widely used products from each reanalysis center: MERRA ANA products, the MERRA-2 ASM file collection, ERA-Interim, JRA-55, and CFSR, with all datasets used on model levels and at the available horizontal resolution closest to the model grid.

## 3.2 Evaluation of Reanalysis Seasonal Climatologies

Figure 7 shows MERRA-2 upper tropospheric jet frequency distributions during the solstice seasons, DJF and JJA, and differences between those and the other reanalyses. Differences in the equinox seasons (not shown) are of similar character, but in general smaller than those shown here. Overall, the differences between MERRA and MERRA-2 are smaller in magnitude than the differences between MERRA-2 and the other reanalyses, which is not surprising given the greater similarity in the models, assimilation systems, and grids used in these related reanalyses. MERRA shows slightly more frequent jets in the re-





gions where they are most persistent (e.g., the NH subtropical jet over Africa and Asia) than in MERRA-2, and at high latitudes (poleward of about 60°) in both hemispheres, with lower jet frequencies in the extratropical regions with moderate to low jet frequencies.

Each of the other reanalyses shows more jets poleward of about 60° latitude in both hemispheres than does MERRA-2 (albeit very slightly in ERA-I). Overall, ERA-Interim shows fewer, and CFSR shows more, midlatitude upper tropospheric jets than does MERRA-2; this general pattern is likely related to the native latitude grid spacing of ERA-I being coarser and that of CFSR being finer than that of MERRA-2: The native Gaussian grid spacing for ERA-I is near 0.7°, that of CFSR is near 0.3°, and the MERRA-2 latitude grid spacing is 0.5°. ERA-I does show a slightly stronger or more persistent subtropical jet in the NH in DJF across Africa, Asia, and the western Pacific, and in the SH in JJA from about 45°E eastward to about 120°W. These are the regions where there is a very strong persistent subtropical jet at a nearly constant location, and may also be related to resolution in that the finer grid of MERRA-2 may lead to more accurate placement of jets that are very near a bin edge, thus making the jet frequency distributions appear sharper in ERA-I than in MERRA-2. JRA-55 generally shows fewer jets than MERRA-2 in midlatitudes. CFSR shows more extratropical jets at all latitudes, but the patterns suggest a slight poleward shift relative to MERRA-2 of the SH subtropical jet in JJA around most of the globe.

Many of the largest differences are in the tropics: In DJF, MERRA-2 shows more frequent/persistent jets than any of the other reanalyses near the equator (primarily just south of it) near 150°W to 90°W, in the westerly circulation downstream of the Australian monsoon. ERA-I and JRA-55 also show considerably lower frequencies of tropical easterlies than MERRA-2 in the both the Australian (DJF, ∼90–140°E near equator) and Asian (JJA, ∼40–140°E just north of equator) monsoon regions, as well as somewhat lower frequencies of midlatitude westerlies that bound the polar side of the Asian monsoon circulation. While CFSR and MERRA show weaker equatorial westerlies than MERRA-2 like the other reanalyses, they show slightly stronger Australian monsoon easterlies in DJF; CFSR also shows generally stronger Asian monsoon easterlies, while MERRA shows a dipole pattern that suggests that the Asian monsoon easterlies peak slightly closer to the equator in MERRA. MERRA-2 shows a stronger Atlantic "westerly duct" (e.g., Horinouchi et al., 2000; Homeyer et al., 2011) in DJF, with all other reanalyses showing a center of negative differences just north of the equator near 50–10°W.

Cross-sections comparing the jet frequency distributions in JJA (Figure 8) show differences that are typical for this view. Most striking are the general patterns of alternating differences in all the comparisons except for those between MERRA and MERRA-2. Since MERRA and MERRA-2 use the same vertical model grids, the altitude locations differ only to the extent that the relationships between pressure and geopotential height (which is converted to geometric altitude) differ, and thus are expected to be much closer to each other than to the levels used in any of the other reanalyses. The primary differences between MERRA and MERRA-2 are an altitude shift in preferred location of the tropical and the SH jets, especially in high latitudes, and higher jet frequencies in MERRA in the high latitude NH. The DJF differences (not shown) are similar, but with a downward shift in MERRA versus MERRA-2 also apparent around the NH subtropical jet, and an opposite shift of the tropical jets (indicating different behavior for the Asian and Australian monsoons, as was seen in the maps).

As shown in Fujiwara et al. (2017) (their Figure 3), all of the reanalyses have vertical spacing finer than 1 km up to about 8 km, where the MERRA/MERRA-2 spacing quickly jumps to about 1.2 km, while that of the others increases gradually





to 1 km at about 14 km, and exceeds that of MERRA/MERRA-2 at about 16 km. Thus, in the altitude region of the strong subtropical jets (11–12 km), ERA-I, JRA-55, and CFSR all have finer vertical spacing than MERRA and MERRA-2, and all show similar patterns of differences, with higher frequencies near the upper part of the subtropical jet surrounded by lower frequencies. For the high latitude jets, the patterns are more complex, but consistent with the differences seen in the maps.

In both zonal mean/altitude and map views including all altitudes, the differences seen here are nearly all less than about 10% of the maxima in the frequency distributions, thus amounting to under 20% of the local frequencies except in regions where jets are very uncommon and in the Asian summer monsoon region. (Recall that, as described in Section 2.3, since frequency is expressed as a percent, the absolute differences between MERRA-2 and other reanalysis frequency distributions are also in units of percent; the relative (percent) differences noted here are obtained by dividing the number in the difference plot

by the number in the MERRA-2 frequency distribution plot.) Differences near the 20% level are much more common in the vertical distribution than in the maps, with only the equatorial circulations showing differences this large in the maps. These differences, albeit substantial, are generally either very localized, suggesting small shifts in the identified positions of the jets, or quite broad, suggesting an overall bias in the number of jets. Given these patterns of differences, the picture of the relative jet frequencies as a function of geographic location is very similar in all of the reanalyses.

Figures 9 and 10 show the differences in multiple tropopause frequencies among the reanalyses. The overall spread among the analyses is considerably larger than that seen for the upper tropospheric jets, with differences of up to about 50% in regions of high multiple tropopause frequencies. As with the jets, differences between MERRA and MERRA-2 are usually less than those between MERRA-2 and the other reanalyses. MERRA shows almost uniformly slightly fewer multiple tropopauses than MERRA-2 in DJF; in JJA, there are larger differences (up to  30% of the corresponding frequency) in the SH, with a nearly

zonally symmetric pattern of fewer SH multiple tropopauses in midlatitudes and more multiple tropopauses in high latitudes.

In DJF, ERA-I shows fewer multiple tropopauses than MERRA-2 near 30° latitude in both hemispheres, and more at higher latitudes. These differences are largest (up to about 30%) in the regions of the westerly ducts and the westerlies of the Walker circulation. JRA-55 has fewer multiple tropopauses than MERRA-2, with largest differences in DJF in the subtropics in both hemispheres, indicating that the multiple tropopauses associated with the temperature structure of the NH lower stratospheric vortex and subvortex in winter are similarly represented in JRA-55 and in MERRA-2. In JJA, largest differences (up to $\sim$30–

40%) are in the SH, from the subtropics to about 65°S. CFSR shows many more multiple tropopauses than any of the other reanalyses globally, with differences from MERRA-2 of 30–50% in midlatitudes and SH winter high latitudes. CFSR also shows a significant number of multiple tropopauses identified in the tropics, which are not present in any of the other reanalyses, and which are especially prominent along the equator in the longitude region of the Asian summer monsoon during JJA.

The cross-sections in Figure 10 indicate that the primary tropopauses are typically near the same altitude in all reanalyses, with latitudinal differences reflecting those seen in the maps. The secondary tropopauses, however, show quite different distributions in different reanalyses in the SH, with MERRA-2 generally showing a distribution that is more localized in the vertical than that of the other reanalyses: There is thus a deficit of multiple tropopauses near 15–17 km in all other reanalyses (including MERRA) with respect to MERRA-2, flanked by regions with more secondary tropopauses above and below. Multiple

tropopauses identified in the polar winter, especially in the SH, are largely a consequence of weak vertical temperature gradients





over a large altitude region, which result in "re-crossing" the lapse rate, and are very sensitive to the details of that temperature structure (Manney et al., 2014; Schwartz et al., 2015, and references therein). The systematic difference in structure between MERRA-2 and the other reanalyses appears broadly consistent with differences in zonal mean temperature structure (e.g., Long et al., 2017). Examination of multiple tropopause differences for the earliest and latest ten years studied here indicates that the

pattern of differences in secondary tropopause altitude between MERRA-2 and the other reanalyses in the climatology is driven primarily by differences in the early years, with recent years showing smaller differences without the consistent high-low-high pattern seen in the climatology. Evaluations of zonal mean temperature structure for S-RIP (summarized by Long et al., 2017) indicate greatly improved agreement in stratospheric temperatures after 1998, when the transition between TOVS and ATOVS radiances was made. Furthermore, polar vortex temperature diagnostics (similar to those in Lawrence et al., 2015, not shown)

an abrupt increase in the agreement between reanalyses in the SH vortex at that time that extends down to at least ∼15 km, below the level of most secondary tropopauses. Together, these suggest that reanalysis temperature differences related to handling of the coarser-resolution TOVS radiances before 1999 is a significant factor in the patterns of SH polar winter multiple tropopause differences seen here. Detailed evaluations of vertical temperature variations in each of these reanalyses, and their impact on multiple tropopause distributions, is work in progress that is beyond the scope of this paper.

In the NH winter (not shown), as is the case in JJA, the primary tropopause altitudes, as well as mid- and low-latitude secondary tropopause altitudes, agree well, with differences reflecting the latitudinal patterns seen in the maps. The secondary tropopause altitudes again differ among the reanalyses, but here it is not a consistent shift with respect to MERRA-2, and significant differences are generally limited to the highest latitudes (poleward of about 70°). These smaller NH differences also appear broadly consistent with the results of Long et al. (2017) and with much smaller differences in temperature diagnostics

in the lowest part of the stratospheric vortex.

Figure 11 shows differences among the reanalyses in subvortex jets in NH winter (DJF). Overall, these differences are small. In the preferred region for subvortex jets (highest frequencies in MERRA-2 plots), they show a high bias in ERA-I and a low bias in JRA-55; in the regions where subvortex jets persist at lower latitudes, especially over eastern Asia and the eastern Pacific, the opposite bias is seen, with ERA-I showing fewer and JRA-55 more subvortex jets than in MERRA-2. Differences

between MERRA-2 and MERRA and CFSR are smaller than those for ERA-I and JRA-55, with slightly larger maximum frequencies in MERRA than in MERRA-2, and slightly lower maximum frequencies in CFSR. The patterns of merged jets (subvortex jets that merge into an upper tropospheric jet at the bottom; not shown) and differences in them among reanalyses are very similar to those shown here for all subvortex jets.

The SH winter (JJA, Figure 12) subvortex jets show a similar picture (again, the results are very similar for merged subvortex

and upper tropospheric jets, not shown). A slight poleward shift is seen in the preferred position of subvortex jets in MERRA with respect to that in MERRA-2, while ERA-I shows a slight equatorward shift with respect to MERRA-2. JRA-55 shows fewer subvortex jets near the preferred region for them, and more at both higher and lower latitudes, suggesting more variability in their locations, but indicating an equatorward shift with respect to MERRA-2 from about 45°W to 90°E longitude. CFSR shows a pattern that is more complex and longitude-dependent, but suggests a poleward shift of the preferred region with

respect to MERRA-2. The zonal mean cross-sections show very small differences between MERRA and MERRA-2 (which





have the same vertical grids). The other reanalyses show patterns of differences that are consistent with the vertical grids. ERA-I and JRA-55 have very similar vertical grids, with finer spacing than the ∼1.2 km MERRA-2 interval below about 16 km and slightly coarser spacing (up to ∼1.4 km) above, while the CFSR resolution remains finer than that of MERRA-2 throughout the region shown (∼0.8–1.0 km) (Fujiwara et al., 2017, Figure 3). Consistent with this, and the patterns seen in the maps, ERA-I and JRA-55 show very similar patterns, with regions of higher and lower frequencies dependent on the relative spacing of the vertical grids and bins. For both ERA-I and JRA-55, lower frequencies than MERRA-2 occur over broader latitude regions than do higher frequencies, suggesting that both of these reanalyses often have lower windspeeds in the lowermost stratosphere than MERRA-2. Conversely, CFSR shows considerably higher integrated frequencies consistent with higher overall windspeeds.

### 3.3 Evaluation of Reanalysis Climatological Annual Cycle

To complement the seasonal snapshots, we show here the climatological annual cycle in the frequency distributions and, for the jets, the associated windspeeds. These are shown for daily values averaged over the 35-year period; thus, while somewhat noisy, they reflect the full degree of scatter and variability in these fields.

Figure 13 shows the frequency distributions and windspeeds for the upper tropospheric jets. While the frequencies evolve through the seasons (as described in detail by Manney et al., 2014), the patterns of differences are quite consistent: MERRA-2 has fewer (and weaker, as seen in the windspeed differences) jets at high latitude than the other reanalyses. ERA-I shows fewer jets than MERRA-2 throughout the domain and year except near each pole (where jet frequencies and windspeeds are both slighter higher than those in MERRA-2), and at the maxima of the frequency distributions. Largest negative differences in the NH subtropical jet are seen in April and May, suggesting that the ERA-I subtropical jet weakens earlier in spring than that in MERRA-2; negative differences in the subtropical jet increase again in October to November, suggesting that the ERA-I subtropical jet also strengthens later in fall. JRA-55 shows similar patterns to ERA-I, including indications that the NH subtropical jet weakens earlier in spring; positive differences near the poles and in much of the SH are larger than those in ERA-I. CFSR shows more jets than MERRA-2 except in low frequency regions in the tropics; the uniformity of the differences throughout the year suggests an overall bias rather than differences in the time of strengthening or weakening. Lower windspeeds are closely correlated with fewer jets except in regions of high windspeeds, where few of the jets are near the threshold value of 40 m/s. The windspeed differences are quite small, usually within ±3 m/s (with these maximum values in regions where the differences exceed the range of the color bar). That such small differences in windspeed lead to significant differences in the jet cores identified highlights the sensitivity of threshold diagnostics such as the jet locations; such diagnostics are, however, widely used because of their value in describing/understanding atmospheric processes.

Figure 14 summarizes how the upper tropospheric jet frequencies and windspeeds are related. The MERRA-2 distribution of frequency versus windspeed constructed from the values in Figure 13 shows peaks in the jet frequency distribution near 45 m/s and 60 m/s. The latter peak arises primarily from the strong jets that persist with nearly constant locations through winter in each hemisphere, while the former reflects the more variable jets in summer and highly variable regions such as over North America, as well as the tropical westerly and easterly jets (which have lower windspeeds). Very similar patterns appear in the other reanalyses (not shown), with the slopes of the linear fits ranging from 0.530 (ERA-I) to 0.564 (CFSR), and the correlation





coefficients from 0.850 (ERA-I) to 0.865 (JRA-55). If differences in windspeeds were the primary reason for the differences in jet frequencies among the reanalyses, we would expect the difference correlation plots to peak at negative/positive frequencies and negative/positive windspeeds. This pattern is seen clearly for ERA-I (third row in Figure 14), where weaker jet windspeeds in ERA-I correspond closely to lower frequencies, that differences in jet frequencies between ERA-I and MERRA-2 arise

largely from lower peak windspeeds in ERA-I is consistent with the results shown previously and with the coarser resolution of ERA-I. MERRA and JRA-55 show a less distinct pattern of this sort, suggesting that some of the differences arise from typically weaker peak windspeeds in those two reanalyses than in MERRA-2; it is not so clear in these cases whether the weaker windspeeds are related to resolution, since MERRA has the same latitudinal and only slightly coarser longitudinal resolution than MERRA-2, and JRA-55 only slightly coarser latitudinal and longitudinal resolution. The CFSR comparison

shows higher frequencies than MERRA-2 that are nearly independent of windspeed, suggesting that windspeed differences are not the primary reason for the frequency differences.

The multiple tropopause frequencies (Figure 15) show lower values in the preferred region along the subtropical jet in both hemispheres in MERRA, ERA-I, and JRA-55 than in MERRA-2 throughout the year. As shown in the maps, CFSR has higher multiple tropopause frequencies globally, consistent with Figure 9. The magnitude of the differences between all reanalyses is

largest in the SH winter, with MERRA and ERA-I showing positive differences near the pole that persist into November. Large, but gradually decreasing, negative differences in SH midlatitudes (about 30° to 60°S) during May through August in MERRA, ERA-I, and JRA-55 suggest that multiple tropopauses in MERRA-2 form later in this region. While systematic differences are seen in the altitude of the primary tropopause, the magnitude of these is typically no more than about 0.5–0.7 km, which is consistent with differences arising from slightly different spacing of model levels that are ∼0.8 to 1.2 km apart at these

altitudes. Primary tropopause altitude differences are not very meaningful in the tropics where multiple tropopause frequencies are very low.

The annual cycle of merged subvortex jet frequencies is shown in Figure 16; the differences in total subvortex jet frequencies (not shown) have very similar patterns to those for the merged jet. Small differences are seen throughout the winter seasons in each hemisphere that are consistent with those seen in maps (e.g., Figures 11 and 12), with ERA-I showing lower frequencies at

low latitudes and higher frequencies at high latitudes in NH winter, and JRA-55 showing the opposite. Differences are typically no more than about 10% of the frequency in MERRA-2. The minimum altitudes of the merged jets are very close in MERRA and MERRA-2, consistent with the use of the same vertical grids. Other reanalyses show differences in minimum altitude that can exceed 2 km, with ERA-I generally having higher minimum altitudes and JRA-55 lower ones, and CFSR showing latitudinally and seasonally varying biases. The differences are generally largest in SH spring and NH fall.

**4   Summary and Conclusions**

We have compared the climatologies of upper tropospheric jets, multiple tropopauses, and subvortex jets in the five latest generation high-resolution reanalyses, for the 35-year period spanning 1980 through 2014. While overall qualitative agreement





is very good, significant quantitative differences illuminate the limits and uncertainties of these reanalyses for UTLS dynamical studies (which in turn have implications for transport and composition of radiatively active trace gases in the UTLS).

Comparisons of occurrence frequency distributions of jets and tropopauses of each of the other reanalyses were made against those in MERRA-2, the most recent of the full-input reanalyses to be released. The other analyses are MERRA, ERA-Interim,

JRA-55, and NCEP's CFSR. Comparisons of different data products from each of these centers highlight some of the sensitivities of the representation of UTLS dynamics to model and data assimilation configuration:

- The MERRA-2 "ANA" (before incremental analysis update) and "ASM" fields show small differences (typically less than 5% for jet frequency distributions and less than 10% for tropopause characteristics) that are nevertheless significant in some regions. For most analyses, including the current work, the ASM fields are recommended as providing the most

complete and dynamically consistent products.

- Differences between the newly available model level CSFR products and those interpolated to a coarser pressure-level grid illustrate the importance of vertical resolution/grid spacing for UTLS analyses. While differences are, as expected, largest for multiple tropopause distributions (up to ∼60%), significant discrepancies (commonly 15 to 30%) are also seen in upper tropospheric jets, and in merged upper tropospheric and subvortex jets.

- Comparison of JMA's JRA-55 with its conventional data only counterpart, JRA-55C, reveals quite small differences in the NH and large differences in the SH, reflecting the sparsity of conventional data in the SH. The largest differences are in high-latitude SH fall and winter multiple tropopauses, which show a dipole pattern in longitude of higher/lower frequencies in JRA-55C poleward of about 65°S (resulting in 20–30% differences in frequencies), and fewer multiple tropopauses around the globe near 40–60°S.

Comparisons of jets and multiple tropopauses in each of the other reanalyses with those in MERRA-2 reveal the following systematic differences:

**Upper tropospheric jet frequency distributions** are generally lower in MERRA, ERA-I, and JRA-55 than in MERRA-2 and generally higher in CFSR. In the polar regions, however, MERRA-2 shows lower frequencies than any of the other reanalyses. Tropical jets associated with the Walker circulation westerlies in NH winter are less frequent/persistent in all of the other

reanalyses than in MERRA-2; Asian and Australian monsoon easterlies are less frequent in ERA-I and JRA-55, and more frequent in CFSR. Monsoon differences between MERRA and MERRA-2 are more complicated, with a stronger Australian monsoon and shift in position/size of the Asian monsoon. Differences in upper tropospheric jet altitude are consistent with the differences in assimilation model vertical grids, with ERA-I and JRA-55 (which have very similar vertical grids) showing more jets than MERRA-2 near the subtropical jet maximum and fewer above and below. A strong upward shift in high-latitude jets

in MERRA-2 versus MERRA is seen in SH winter.

**Multiple tropopause frequency distributions** indicate fewer globally in ERA-I and JRA-55 than in MERRA-2, and more in CFSR. As for the upper tropospheric jets, the only significant differences between MERRA and MERRA-2 are in the SH winter in middle to high latitudes. Primary tropopause altitudes are similar in all reanalyses, but secondary tropopause altitudes



in the SH in MERRA-2 are more clustered at the same altitude than in the other reanalyses. CFSR shows many more multiple tropopauses in the tropics than the other reanalyses.

**Subvortex jet frequency distributions** show relatively small differences among the reanalyses. ERA-I shows slightly higher, and JRA-55 slightly lower, maximum subvortex jet frequencies in NH winter, while MERRA / MERRA-2 NH winter dif-

5 ferences are nearly negligible. CFSR / MERRA-2 differences are also very small in NH winter. In SH winter, differences in geographic existence patterns are again small, with slight latitude shifts indicated in MERRA, ERA-I, and CFSR, and a less sharply peaked pattern in JRA-55 than in MERRA-2. Vertical distributions show patterns related primarily to the differing vertical grids.

In general, the reanalyses show modest quantitative differences in the distributions of UTLS jets and multiple tropopauses,

most of which are consistent with expectations based on differences in assimilation model grids and resolution. ERA-I typically shows a significant low bias in upper tropospheric jets with respect to MERRA-2 and JRA-55 a more modest one, while CFSR shows a high bias in both upper tropospheric jets and multiple tropopauses. With a few exceptions, differences between MERRA and MERRA-2 are very small. These patterns of frequency differences may arise partially from the fact that ERA-I has coarser and CFSR finer native horizontal resolution than MERRA and MERRA-2 – for these threshold phenomena, a

finer grid is likely to more accurately pinpoint the location where that threshold is crossed, particularly in the case of upper tropospheric jets, for which the criterion is a single maximum in the latitude/altitude plane. For multiple tropopauses, the vertical grid spacing and details of vertical temperature structure are particularly critical.

The only places where MERRA and MERRA-2 show substantial differences are in the mid- to high-latitude SH winter upper tropospheric jets and multiple tropopauses, and in the upper tropospheric jets associated with the tropical circulations during

the solstice seasons. These are also times and places where some of the largest differences from the other reanalyses are seen. The MERRA/MERRA-2 difference in multiple tropopauses are more pronounced in the earliest decade of the comparison than in the latest, suggesting that they arise from differences in temperature structure (as reported in zonal mean fields by Long et al., 2017) related to changes in the satellite radiance inputs to the reanalyses. Note that another difference between MERRA-2 and the other reanalyses is its assimilation of MLS and OMI ozone data in a system where assimilated ozone is interactive with the

radiation code; in the SH winter and spring this significantly changes the assimilated ozone (Davis et al., 2017; Wargan et al., 2017); whether significant differences in temperature structure may arise from this is a subject for future exploration. Coy et al. (2016) showed improved representation of the Quasi-Biennial Oscillation in MERRA-2 versus MERRA (in part because of improvements in the equatorial gravity wave drag parameterization) that likely reflects a general improvement in capturing tropical circulations.

The differences overall show very good qualitative agreement among the reanalyses, giving high confidence in the large scale climatological features of the UTLS jet and multiple tropopause distributions. Figure 17 shows that, for most fields compared here, in the largest scale global picture, the reanalyses agree quite well quantitatively; especially in the case of the jets (upper tropospheric and subvortex), this likely reflects the overall similar and accurate representation of large scale dynamics in all of the models and the first-order effects of assimilating largely the same datasets. This view is supported by the fact that (in

contrast to the situation for multiple tropopauses) examination of the first and last decades of the comparison period (not





shown) indicates no substantial changes in the upper tropospheric and subvortex jet differences. As noted above, and seen in the second row of Figure 17, larger differences are seen globally in multiple tropopause occurrence and altitudes than for the jets, with CFSR showing higher frequencies globally and significant differences in the peak altitude and altitude distributions of the secondary tropopause. The merge altitude of merged upper tropospheric and subvortex jets (Figure 17, bottom right panel)

also shows somewhat larger differences. Multiple tropopause frequencies and altitudes and upper tropospheric/subvortex jet merge altitudes are strongly dependent on vertical resolution and grid spacing, thus differences in reanalysis vertical grids are reflected globally in these fields.

We have shown above that larger quantitative differences are seen on regional and seasonal scales. These differences may have important consequences for the representation or simulation of transport of radiatively active trace gases such as ozone and

water vapor in the UTLS; a follow-on paper will examine assimilated ozone in a jet and tropopause focused framework compared with Aura Microwave Limb Sounder observations as a way of assessing these effects. Because derived quantities such as global locations, distributions, and strength of jets cannot be compared directly with observations, the degree of agreement among state-of-the-art reanalyses is an important tool for assessing uncertainties in our knowledge of their climatology and variability. In a concurrent paper, we use agreement among these reanalyses to assess the robustness of variability and trends

in upper tropospheric jet locations and windspeeds (Manney and Hegglin, 2017). The significance of the choice of which reanalysis or reanalyses to use will depend strongly on the type of study: While the large scale climatological picture seen in each of the reanalyses is very robust, differences in regional and seasonal distributions, especially of multiple tropopauses and tropical upper tropospheric jets, may have significant consequences. Studies relying on these patterns should thus ideally evaluate more than one reanalysis. Because of the importance of resolution and model grids in characterizing UTLS jet and

tropopause structure, assessing the impact of using different reanalyses is particularly critical when assimilated meteorological fields are used to evaluate the representation of UTLS jets and tropopauses in global chemistry-climate models.

## 5   Data Availability

The datasets used are publicly available, as follows:

- MERRA-2: https://disc.sci.gsfc.nasa.gov/uui/datasets?keywords=%22MERRA-2%22

- MERRA: https://disc.sci.gsfc.nasa.gov/uui/datasets?keywords=%22MERRA%22

- ERA-I: http://apps.ecmwf.int/datasets/

- JRA-55: Through NCAR RDA at http://dx.doi.org/10.5065/D6HH6H41

- JRA-55C: Through NCAR RDA at https://doi.org/10.5065/D67H1GNZ

- CFSR, pressure level data: Through NCAR RDA at http://dx.doi.org/10.5065/D69K487J

- CFSR model level data: Available upon request from Karen H Rosenlof (karen.h.rosenlof@noaa.gov)





– JETPAC products: Contact Gloria L Manney (manney@nwra.com)

*Author contributions.* GLM and MIH designed the studies. LFM, GLM, and WHD ran the JETPAC processing and postprocessing. BWK, ZDL, AL, RAF, and WHD obtained, managed, and formatted the reanalysis datasets. KW and SP provided advice and analysis relating to usage and characteristics of MERRA and MERRA-2 reanalyses. ZDL, AL, MIH, MJS, MLS, and LFM provided advice on figure selection,
5  content, layout, and interpretation. GLM wrote the paper. The coauthors read and commented on the manuscript.

*Competing Interests.* The authors declare that they have no conflict of interest.

*Acknowledgements.* We thank the MLS team for computational, data processing, management, and analysis support; Nathaniel Livesey for helpful suggestions; NASA's GMAO, ECMWF, JMA, and NCEP for providing their assimilated data products; and Amy Butler, Jeremiah Sjoberg, Craig Long, Sean Davis, Henry L Miller, and Karen Rosenlof for processing and providing the model level CFSR data. Work
10  at the Jet Propulsion Laboratory, California Institute of Technology, was done under contract with the National Aeronautics and Space Administration.





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



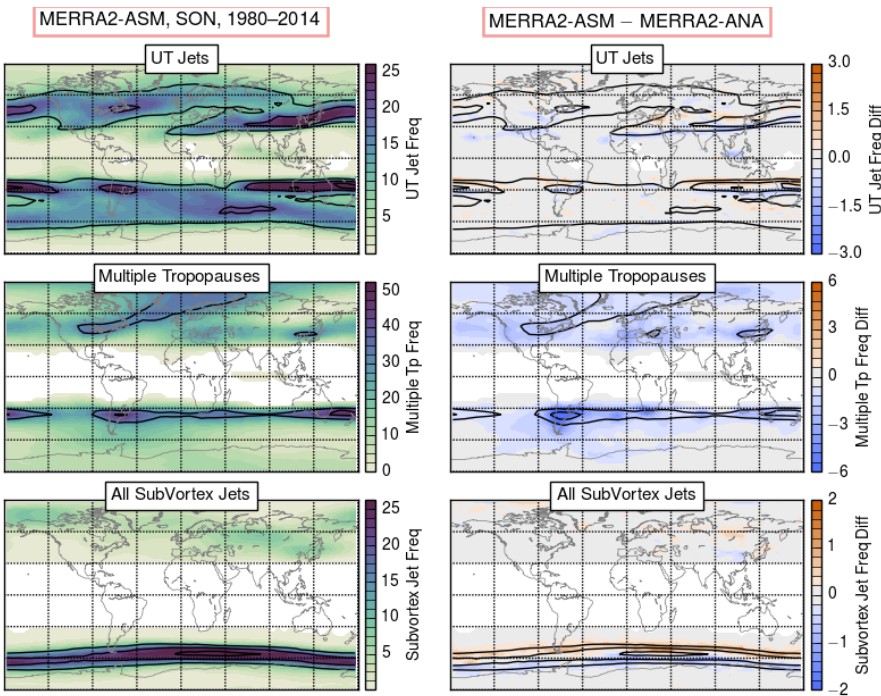

**Figure 1.** Seasonal maps for SON in 1980 through 2014 of MERRA-2 frequency distributions, from ASM (see text; left) fields, and the difference between ASM and ANA (see text). The rows show (top to bottom): upper tropospheric jet frequency, multiple tropopause frequency, and frequency of subvortex jets. Overlaid contours highlight the ASM distributions on the left (ASM plots) and ANA distributions on the right (difference plots). Frequencies are normalized as described in Section 2.3. Overlaid contours show frequency values from each reanalysis of 10, 20, and 30% for upper tropospheric and subvortex jets, and 30, 45, and 60% for multiple tropopauses.



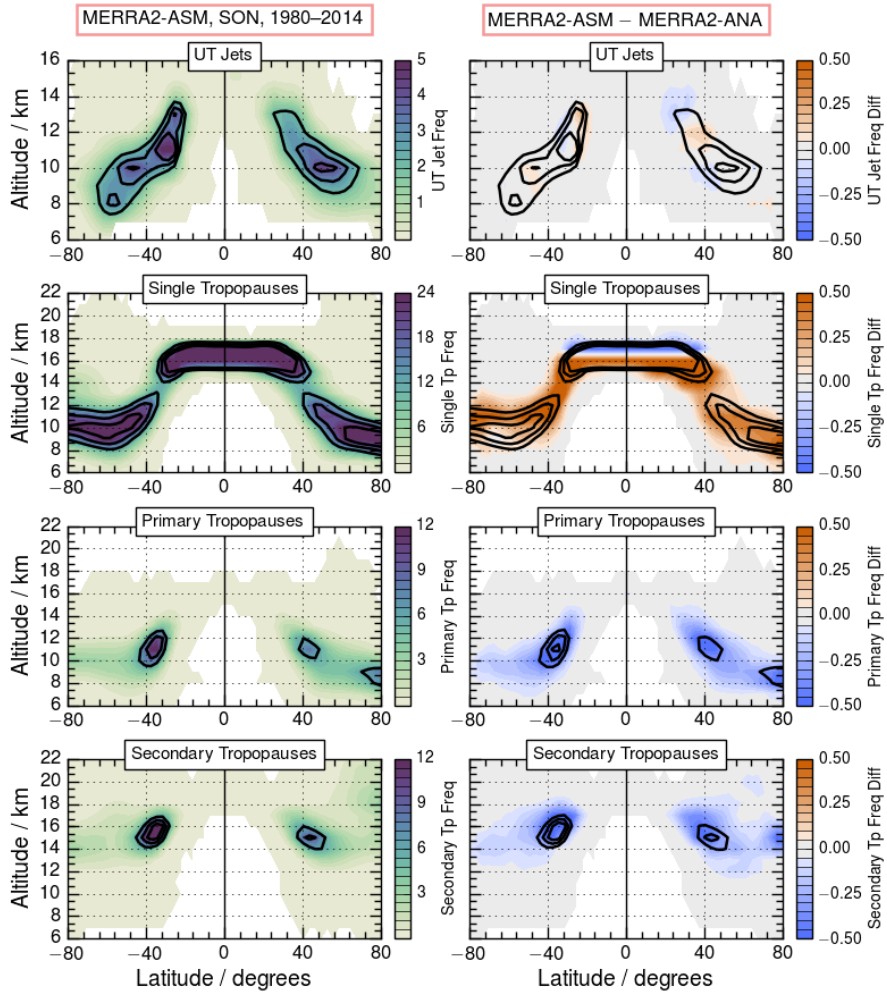

**Figure 2.** Seasonal cross-sections for SON in 1980 through 2014 of MERRA-2 frequency distributions, from ASM (left) and the difference between ANA and ASM (right). The rows show (top to bottom): upper tropospheric jet frequency, single tropopause frequency, frequency of primary multiple tropopause, and frequency of secondary multiple tropopause. Overlaid contours highlight the ASM distributions on the left and ANA distributions on the right. Frequencies are normalized as described in Section 2.3. Overlaid contours show frequency values from each reanalysis of 2, 3, and 4% for upper tropospheric jets and 12, 18, and 24% for multiple tropopauses.



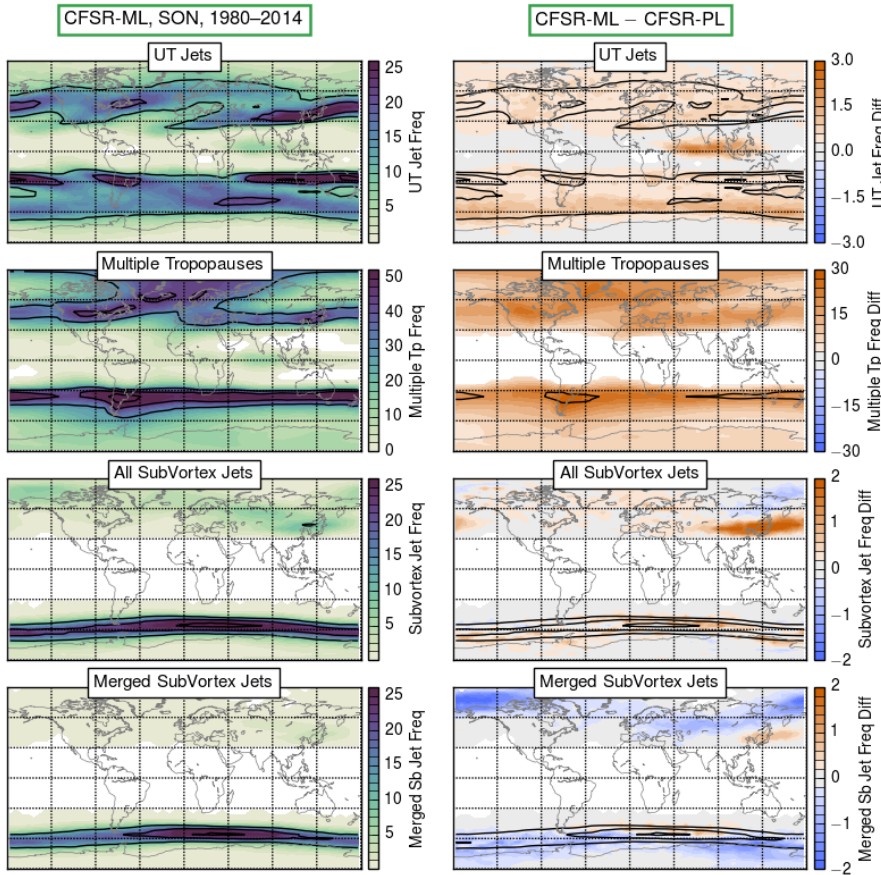

**Figure 3.** Seasonal maps for SON in 1980 through 2014 of CFSR frequency distributions, from model level data (left) and the difference between model and pressure level data (right). Layout is as in Figure 1, except frequencies of merged subvortex jets are shown in the fourth row.





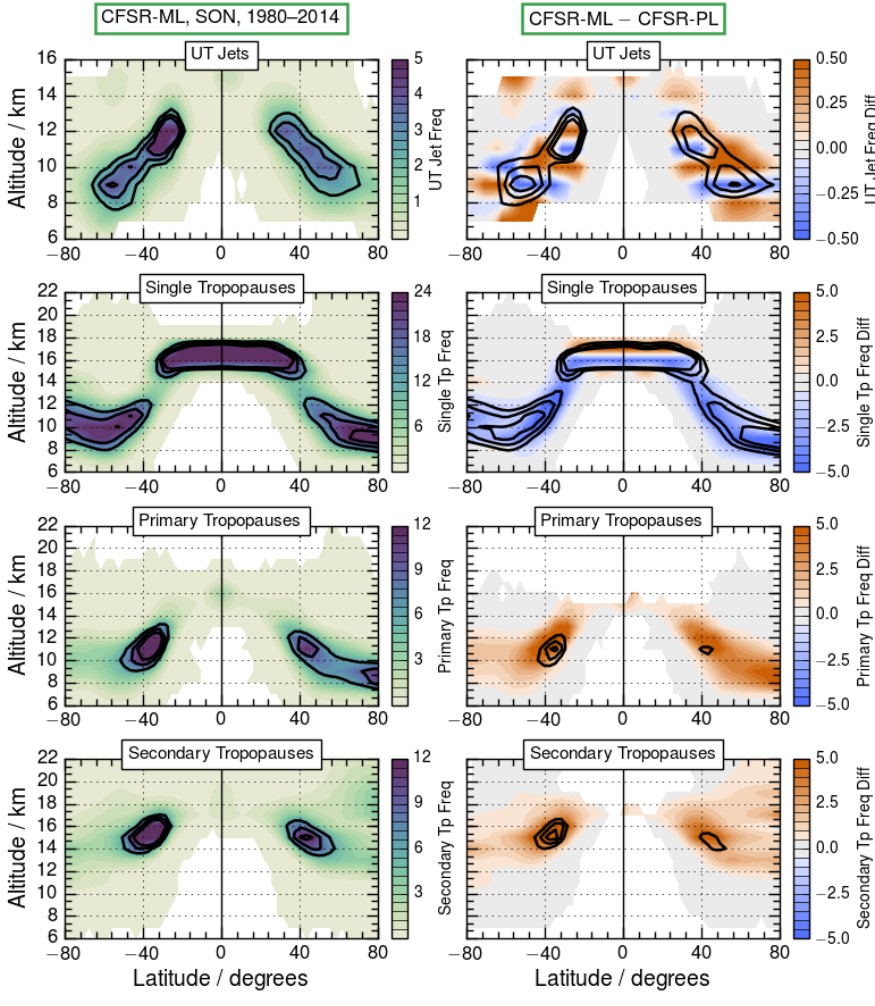

**Figure 4.** Seasonal cross-sections for SON in 1980 through 2014 of CFSR frequency distributions, from model level data (left) and the difference between model and pressure level data (right). Layout is as in Figure 2.



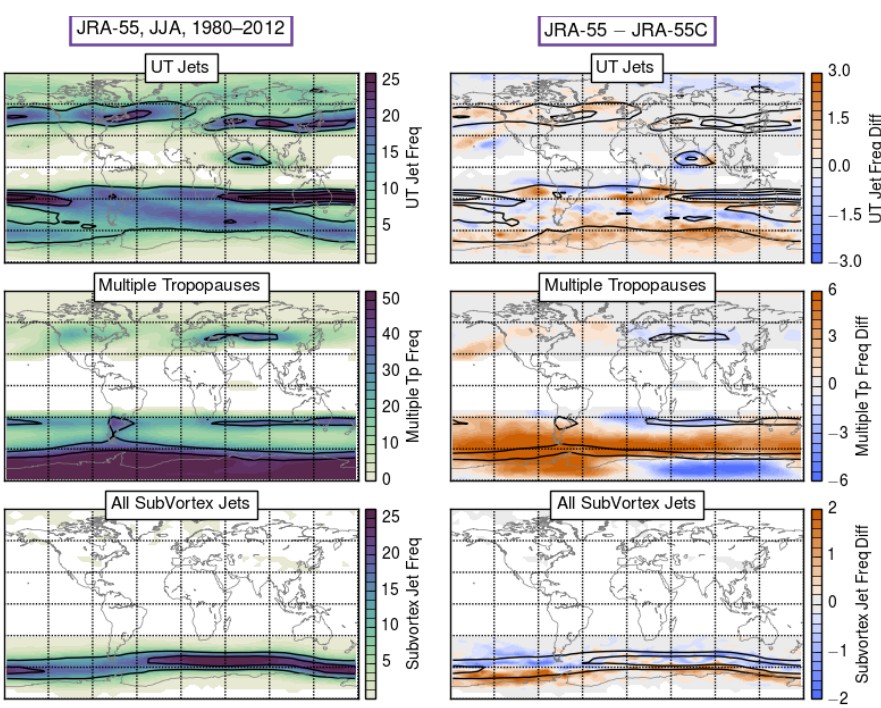

**Figure 5.** Seasonal maps for JJA in 1980 through 2012 of JRA-55 (left) and the difference between JRA-55 and JRA-55C (right). Layout is as in Figure 1.





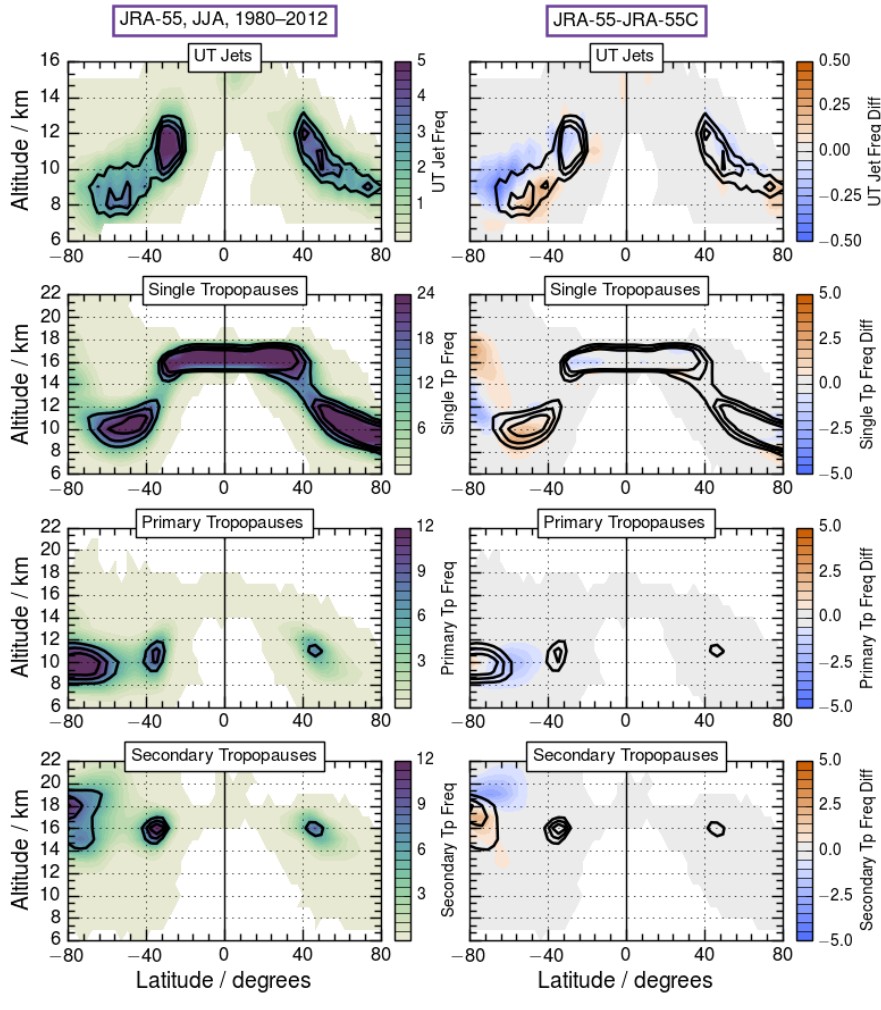

**Figure 6.** Seasonal cross-sections for JJA in 1980 through 2012 of JRA-55 (left) and the difference between JRA-55 and JRA-55C (right). Layout is as in Figure 2.



**Figure 7.** (Left) DJF and (right) JJA maps for 1980 through 2014 of MERRA-2 upper tropospheric jet frequency distributions, and differences between MERRA-2 and MERRA, ERA-I, JRA-55, and CFSR. Overlaid contours are climatological frequency distributions for each reanalysis of 15, 30 and 45%.




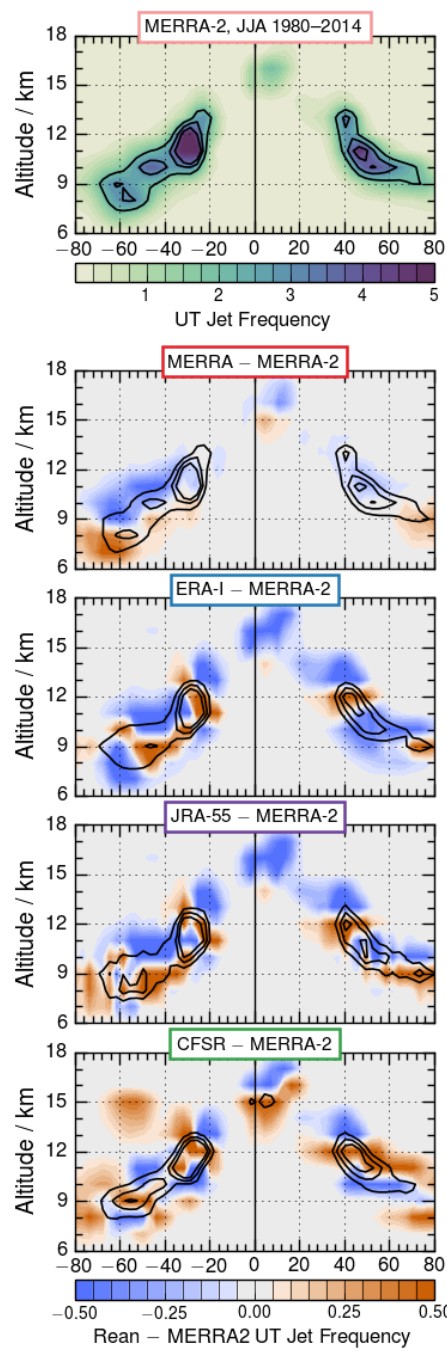

**Figure 8.** Latitude/altitude cross-sections of JJA jet frequency distributions for MERRA-2 (top), and differences between MERRA-2 and MERRA, ERA-I, JRA-55, and CFSR. Overlaid contours are climatological frequency distributions for each reanalysis of 2, 3, and 4%.





**Figure 9.** As in Figure 7, but for multiple tropopause frequency distributions. Overlaid contours show frequency values from each reanalysis of 30, 45, and 60%.




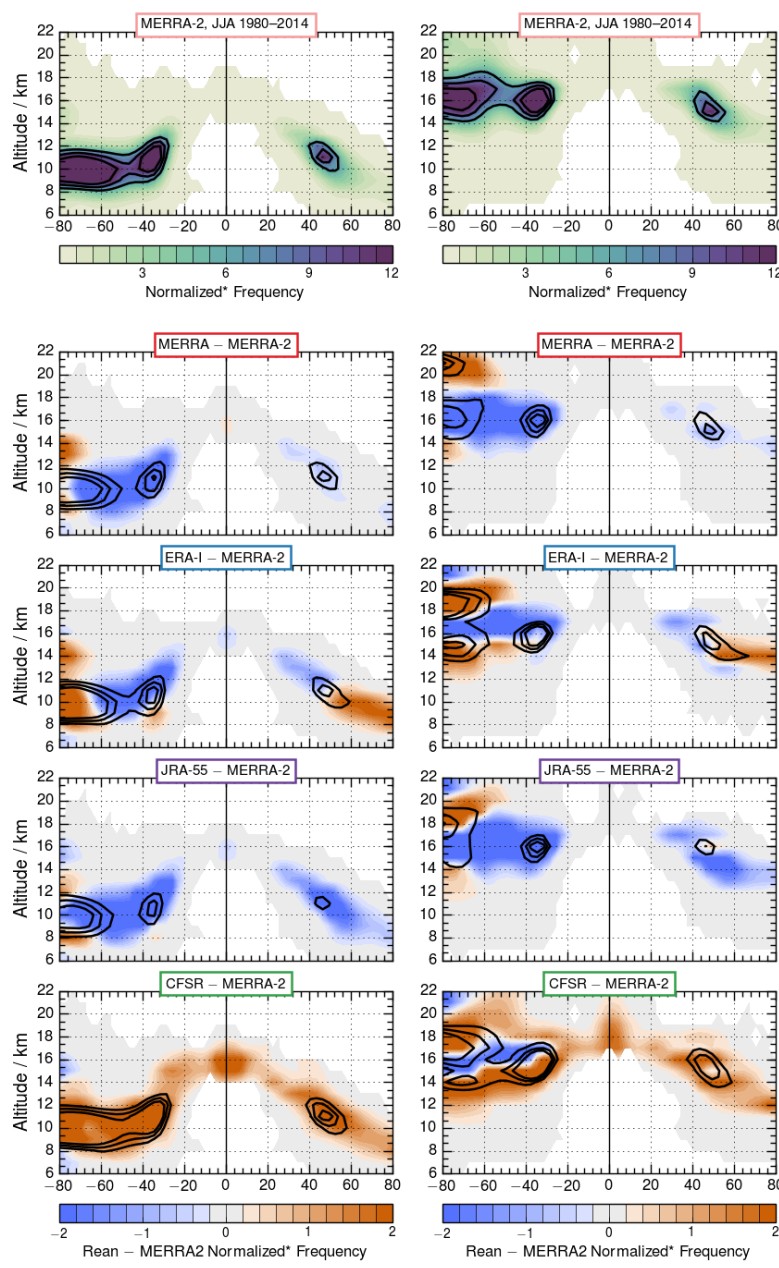

**Figure 10.** As in Figure 8, but for multiple tropopause frequency distributions. Primary tropopause frequencies are shown on the left and secondary tropopause frequencies on the right. Overlaid contours are frequencies of 6, 9, and 12%.





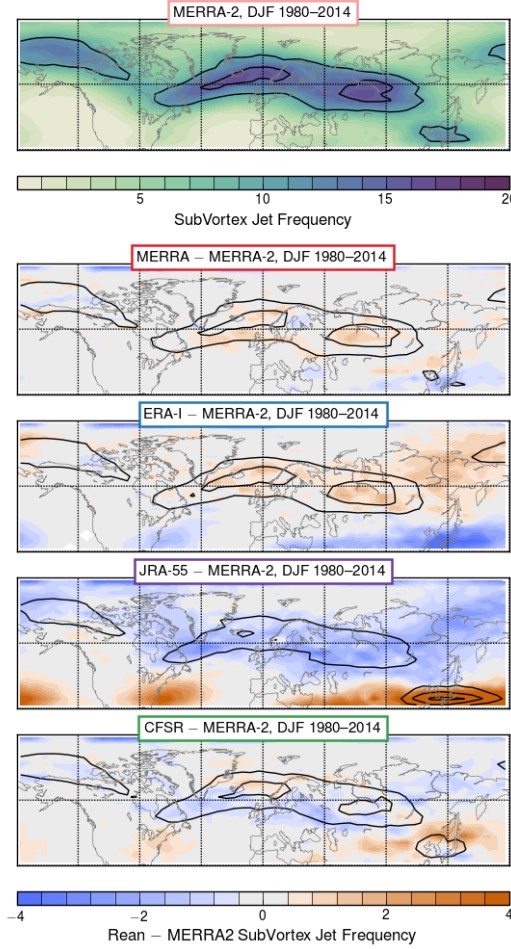

**Figure 11.** As in Figure 7, but for NH subvortex jet frequency distributions in DJF. The latitude domain shown is north of 30°N. Overlaid contours are frequencies of 10, 15, and 20% for each reanalysis.



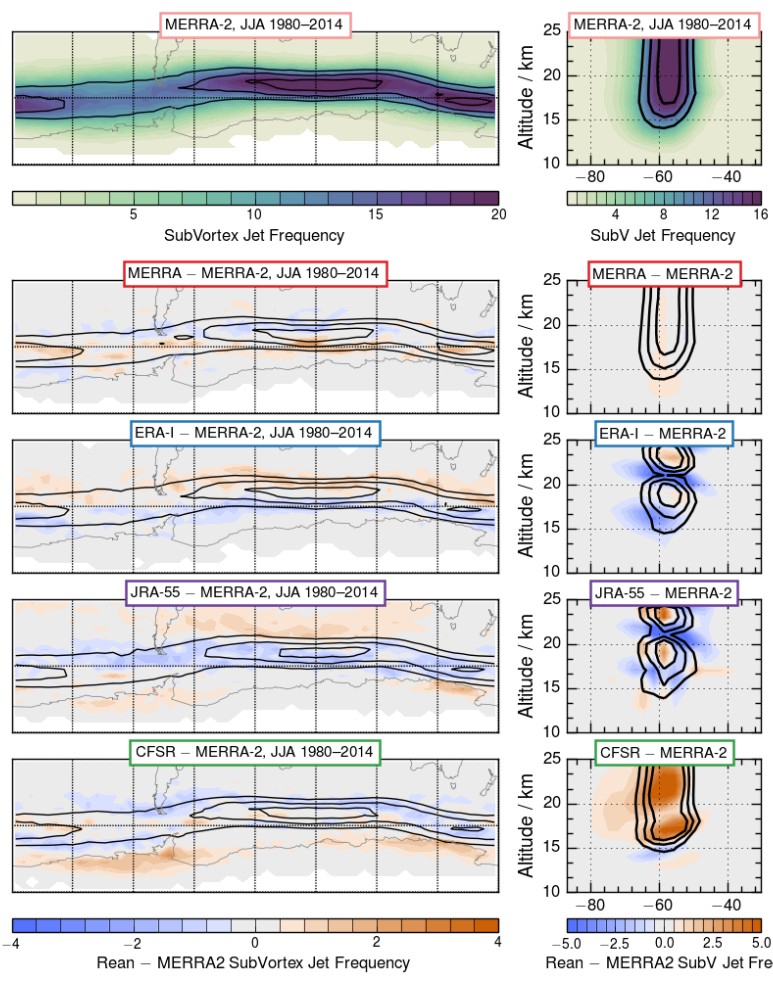

**Figure 12.** As in Figure 11, but for maps (left) and cross-sections (right) of SH frequency distributions of all subvortex jets in JJA. The latitude domain shown is south of 30°S.





**Figure 13.** Climatological seasonal cycle in upper tropospheric jet frequencies for MERRA-2 compared with the other reanalyses. Jet frequency distributions are shown on the left, and mean windspeeds at jet cores on the right. Overlaid contours are climatological values for each reanalysis of 10 and 15% for frequencies and 60 and 72 ms$^{-1}$ for windspeeds.





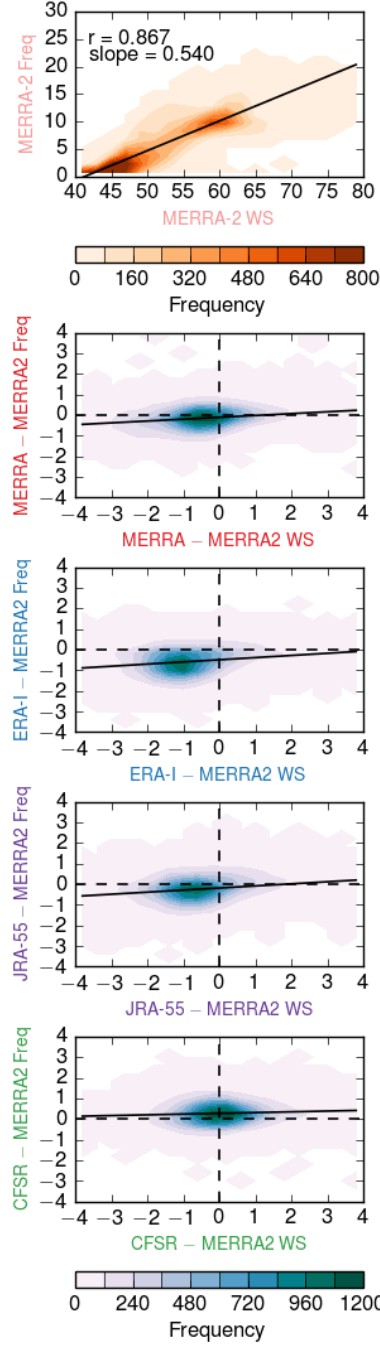

**Figure 14.** Density plots of (top) MERRA-2 climatological daily jet frequencies versus climatological daily jet windspeeds (values are from Figure 13), and of the corresponding (reanalysis − MERRA-2) frequency differences versus jet windspeed differences for (second to fifth rows) MERRA, ERA-I, JRA-55, and CFSR. Black lines are the linear fit to the distributions.





**Figure 15.** As in Figure 13 but for multiple tropopauses, with frequency distributions on the left and primary tropopause altitude on the right. Overlaid contours are 24 and 48% for frequencies and 10 and 14 km for altitudes.





**Figure 16.** As in Figure 13 but for merged subvortex jets, with merge altitude on the right. Overlaid contours are 12 and 16% for frequencies and 14 and 18 km for merge altitudes.



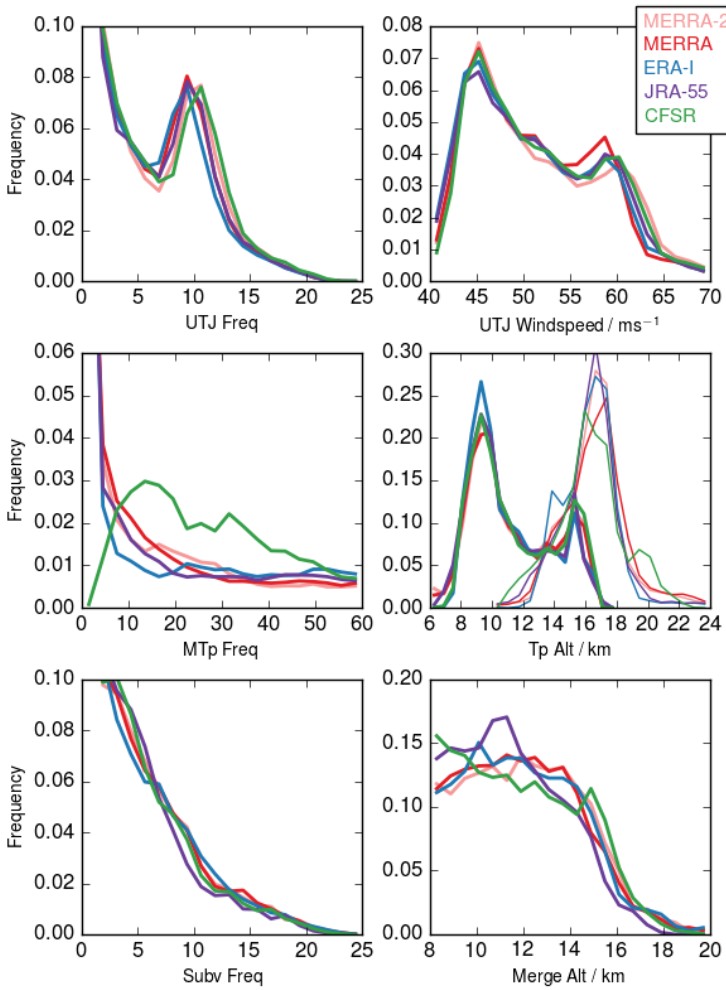

**Figure 17.** Frequency distributions summarizing the global differences in (top row) upper tropospheric jet frequency distributions and wind-speeds, (second row) multiple tropopause distributions and primary (thick lines) and secondary (thin lines) tropopause altitudes, and (bottom row) subvortex jet frequency distribution and merge altitudes for merged subvortex jets. Values summarized are from timeseries such as those in Figures 13, 15, and 16. Pink, red, blue, purple, and green lines show MERRA-2, MERRA, ERA-I, JRA-55, and CFSR, respectively.