# Peer review of "Reanalysis comparisons of upper tropospheric/lower stratospheric jets and multiple tropopauses"

_Atmospheric Chemistry and Physics, 2017_

## Referee Comment (RC1) · Anonymous Referee #2 · 12 Jun 2017

General Comments

In this paper Manney et al. present an intercomparison of UT/LS and stratospheric jets and tropopause diagnostics derived from five recent meteorological reanalyses. The study covers ERA-Interim, JRA-55, MERRA, MERRA-2, and NCEP's CFSR for a 35-year time period (1980-2014). The study found very good qualitative agreement between the climatological features. It is speculated that quantitative differences, which are found to be largely related to differences in resolution of the forecast models and output grids, would still be important for transport and variability studies.

Overall, I found this to be a carefully designed study. The paper is well-written and fits in the scope of ACP. I would recommend it for publication once the few remaining minor comments and technical corrections listed below have been considered.

[Figure]

Specific Comments

p4, l16: It might be worthwhile mentioning some of the updates from MERRA to MERRA-2 that are possibly relevant for this study at this point?

p6, l7-11: I was wondering if the estimation of tropopause heights does involve any kind of higher-order interpolation (e.g. cubic spline) of the coarse-grid temperature profiles from the reanalyses on a fine vertical grid?

p6, l14: Why doesn't it make sense to construct means of frequency distributions from multiple reanalyses?

p6, l17-21 (and other places): It might be more clear if arithmetic differences of percentages would be referred to as "percentage points (pp)" rather than using "%" as the unit for these differences.

p8, l9-16: Can you provide a physical explanation of the northward shift of the jets between the MERRA-2 ASM and ANA fields?

p9, l12-21: Here I was also wondering if you could possibly provide a more detailed physical reasoning for the differences, the jet shifts in particular?

Technical Corrections

p6, l14: doesn't -> does not

p10, l18-19: "in the both the" -> "in both the"

p12, l9-11: A verb seems to be missing in this sentence.

p18, l1: The acronym "JETPAC" was not introduced before.

———————————————

---

## Short Comment (SC1) · 20 Jun 2017

Manney et al. present a comparison of the representation of jets and multiple tropopauses across several reanalysis in the framework of the S-RIP. Therefore itself this paper tries to make comparisons of climatological fields and attribute differences between results through the different reanalysis.

**Major comments:**

- My main concern with this study is about the definition that the authors use for the tropopause and multiple tropopauses. Being clear about this the document to cite in the second paragraph in page 6 is the WMO definition of the tropopause (see below) and it could be that the computation of multiple tropopauses that

the authors have done is wrong, as the WMO definition states that a multiple tropopause needs a vertical thermal gradient of 3 degrees to be considered as such, and not 2 degrees (Celsius not Kelvin according to the original definition to be fair). This should be double checked by the authors and figures replotted if necessary.

World Meteorological Organization (1957), Meteorology: A three dimensional science, WMO Bull., 6, 134-138.

• Other issue is that the paper does not include a proper discussion on how the results here shown compare to the existing literature. This is specially important because this paper deals with reanalysis and previous results include radiosonde or GPS-RO data. Therefore I think that it would be really useful a section discussing the results of multiple tropopauses (at least for the well known planetary hotspots) in comparison with those obtained by Schmidt et al. (2006), Randel et al. (2007) (already cited) and Añel et al. (2008). Again I acknowledge that the focus of the paper is on the intercomparison, but maybe a good idea of doing this is to include in the discussion the spread of the reanalysis respect to the existing literature (e.g. the reanalysis witht the minimum value for MTs over Japan is X with a value of Y and this is in the range (or not) of the values obtained by previous works). Maybe a table for the four hotspots of the North Hemisphere and South Hemisphere would be a good idea.

Schmidt et al. 2006 A climatology of multiple tropopauses derived from GPS radio occultations with CHAMP and SAC-C, Geophys. Res. Lett.

Añel et al. 2008 Climatological features of global multiple tropopause events, J. Geophys. Res.

- Also along the text and figures I have not seen a clear statement on what the authors mean by 'multiple tropopauses'. Multiple tropopauses should not be confound with double tropopauses. Right now there is a pretty clear distinction in the literature about this. From the manuscript I guess that the authors refer to double tropopauses all along the text and not multiple (e.g. triple tropopauses or above). A clear statement of what is considered as multiple tropopause should be included. For example, are you mixing cases of double tropopauses and multiple tropopauses? this could lead to inhomogeneous results because of the vertical resolution of the reanalysis. Anyway a clarification is needed.

Minor comments:

- page 2, line 22: 'they'

- page 2, line 32: Chen et al. 2013 shows a nice case study along three field campaigns with radiosondes, combined with GOME-2 ozone data, lagrangian transport modeling of STE exchange and jet analysis that could be helpful to additionally support this view:

  Chen et al. 2013 The deep atmospheric boundary layer and its significance to the stratosphere and troposphere exchange over the Tibetan Plateau , PLoS ONE

- Along the text the surname 'Peña' is not well written, it would be good to write it correctly with the 'ñ'. It is just necessary copy+paste or with LaTeX to write it as 'ñ'

- page 3, line 20: it would be good to put the 'th' as uppercase

- there are some minor typos along the text, please double check them.

---

## Referee Comment (RC2) · Anonymous Referee #1 · 4 Jul 2017

This paper provides a comprehensive study of the dynamics of the UTLS regions as encapsulated in the jets and multiple tropopauses occurring in this region. The paper provides useful information to the atmospheric sciences community. The discussion of the performance of the reanalyses further provides information on their ability to represent the dynamics of the UTLS region. I recommend acceptance for ACP once the authors address the specific comments below.

Specific comments

P. 5

L. 23 and elsewhere: Is the standard convention ms-1 or m/s? Same for similar ratios

(and see caption for Fig. 13).

P. 6

L. 9-11: Perhaps rephrase, as the meaning of this sentence is not fully clear to this reviewer.

L. 18: As a percent of what?

P. 8

L. 29: Why is this unsurprising?

P. 10

L. 18: Omit superfluous "the".

P. 14

Sect. 4, Summary and conclusions: This is an editorial decision, but I would suggest the authors consider splitting this section into two, a summary and a conclusion sections with the latter being succinct.

P. 23

Fig. 1: Not clear to this reviewer which contour line corresponds to each percentage. I suggest the authors make this clearer in the figure, maybe by attaching the percentage to the contour (same for other figures, e.g., Fig. 2). I also suggest the authors consider providing information in the caption what the colours indicate, e.g., red/blue indicate positive/negative differences. Same for other figures.

P. 28

Fig. 6: The authors should clarify in the caption the differences between Figs. 5 and 6.

---

## Author Comment (AC1) · 7 Jul 2017

*J. A. Añel*

We thank Dr. Añel for his interest and his helpful comments. His comments are shown in blue in italics, and our responses in black.

*Major comments:*

*My main concern with this study is about the definition that the authors use for the tropopause and multiple tropopauses. Being clear about this the document to cite in the second paragraph in page 6 is the WMO definition of the tropopause (see below) and it could be that the computation of multiple tropopauses that the authors have done is wrong, as the WMO definition states that a multiple tropopause needs a vertical thermal gradient of 3 degrees to be considered as such, and not 2 degrees (Celsius not Kelvin according to the original definition to be fair). This should be double checked by the authors and figures replotted if necessary.*

We apologize for not being more explicit about the definition we use in the paper. As is described in detail in Manney et al (2011, 2014), we followed Randel et al (2007), who found that for relatively coarse vertical resolution reanalyses such as these, relaxing the definition such that the vertical thermal gradient for the secondary tropopause was only 2 C / km (or equivalently 2 K/km) resulted in multiple tropopause distributions that were more comparable with those obtained from high-resolution measurements such as those from GPS-RO or radiosondes using the more stringent definition. We will state this explicitly in the revised paper.

*Other issue is that the paper does not include a proper discussion on how the results here shown compare to the existing literature. This is specially important because this paper deals with reanalysis and previous results include radiosonde or GPS-RO data. Therefore I think that it would be really useful a section discussing the results of multiple tropopauses (at least for the well known planetary hotspots) in comparison with those obtained by Schmidt et al. (2006), Randel et al. (2007) (already cited) and Añel et al. (2008). Again I acknowledge that the focus of the paper is on the intercomparison, but maybe a good idea of doing this is to include in the discussion the spread of the reanalysis respect to the existing literature (e.g. the reanalysis witht the minimum value for MTs over Japan is X with a value of Y and this is in the range (or not) of the values obtained by previous works). Maybe a table for the four hotspots of the North Hemisphere and South Hemisphere would be a good idea.*

Using the MERRA reanalysis, Manney et al (2014) provided a detailed climatology of upper tropospheric jets, multiple tropopauses, the subvortex jet, and the relationships between these features. Detailed discussion is given in that paper of the multiple tropopause distributions in relation to those in previous literature, including Randel et al (2007), Añel et al. (2008), and Peevey et al (2012) (see Manney et al, 2014, pages 3257--3259, and the paragraph spanning pages 3263 and 3264); Añel et al (2008) and, especially, Peevey et al (2012) in turn contain

detailed comparisons of multiple tropopause distributions with previous studies. Given the extensive comparisons already in the literature, including Manney et al (2014) with one of the reanalyses used here and with exactly the same methods, and the fact that the focus is on intercomparison, we do not feel it is important to include further detailed comparisons with previous studies here. It is difficult, if not impossible, to quantitatively compare these studies because of the vast differences in geographic sampling and methods (bin size, normalization, etc) used to construct the frequency distributions; Manney et al (2014) contains some discussion of spatial and temporal sampling effects that may result in some of the differences seen between results from previous studies.

A comparison of multiple tropopauses in these reanalyses with those derived from other datasets such as GPS-RO, radiosondes, or relatively high-resolution satellite datasets such as HIRDLS, if done using consistent methods across all datasets and accurately accounting for geographic sampling differences, would be a valuable study but is well beyond the scope of this paper.

We will, however, add a paragraph explicitly referring to these previous comparisons, and giving the "spread" among the reanalyses considered here.

*Also along the text and figures I have not seen a clear statement on what the authors mean by 'multiple tropopauses'. Multiple tropopauses should not be confound with double tropopauses. Right now there is a pretty clear distinction in the literature about this. From the manuscript I guess that the authors refer to double tropopauses all along the text and not multiple (e.g. triple tropopauses or above). A clear statement of what is considered as multiple tropopause should be included. For example, are you mixing cases of double tropopauses and multiple tropopauses? this could lead to inhomogeneous results because of the vertical resolution of the reanalysis. Anyway a clarification is needed.*

As discussed in Manney et al (2011, 2014) and Schwartz et al (2015), the term "multiple tropopause" is used to mean any profile with more than one tropopause, and "double tropopause" to mean profiles with exactly two tropopauses. These definitions are consistent with those that we have seen used in the extensive literature on double and multiple tropopauses. Thus, as quantified by Schwartz et al (2015), there are a very small fraction of profiles included in this analysis that have more than two tropopauses. Including or excluding these does not change our results in any significant respect. We will add a note to this effect in the methods description in the revised paper.

*Minor comments:*

*page 2, line 22: 'they'*

When a colon is used to divide independent clauses, and there are two or more sentences that follow from the clause preceding the colon (as is the case here), the word following the colon is capitalized.

*page 2, line 32: Chen et al. 2013 shows a nice case study along three field campaigns with radiosondes, combined with GOME-2 ozone data, lagrangian transport modeling of STE exchange and jet analysis that could be helpful to additionally support this view:*

Thank you for this citation -- we will include it with the others in support of this point.

*Along the text the surname 'Peña' is not well written, it would be good to write it correctly with the 'ñ'. It is just necessary copy+paste or with LaTeX to write it as 'ñ'*

We apologize for this oversight; it has been corrected.

*page 3, line 20: it would be good to put the 'th' as uppercase*

In the references I find to this reanalysis (e.g., the online information on it at NOAA), the "th" is not capitalized.

*there are some minor typos along the text, please double check them.*

We will, indeed, proofread the revised paper very carefully, and already have some corrections pointed out by the referees.

---

## Author Comment (AC2) · 4 Aug 2017

*General Comments*

*In this paper Manney et al. present an intercomparison of UT/LS and stratospheric jets and tropopause diagnostics derived from five recent meteorological reanalyses. The study covers ERA-Interim, JRA-55, MERRA, MERRA-2, and NCEP's CFSR for a 35-year time period (1980-2014). The study found very good qualitative agreement between the climatological features. It is speculated that quantitative differences, which are found to be largely related to differences in resolution of the forecast models and output grids, would still be important for transport and variability studies. Overall, I found this to be a carefully designed study. The paper is well-written and fits in the scope of ACP. I would recommend it for publication once the few remaining minor comments and technical corrections listed below have been considered.*

We thank the referee for their helpful comments. The referee's comments are shown in blue italics and our responses in black. (Note that in addition to addressing the referees' comments, the figures have been re-done to use the recently updated "standard" S-RIP colors for each reanalysis.)

*Specific Comments*

*p4, l16: It might be worthwhile mentioning some of the updates from MERRA to MERRA-2 that are possibly relevant for this study at this point?*

We have added this to the text as follows:
"MERRA-2 (Gelaro et al., 2017) uses a similar model and assimilation system to MERRA, with updates also described by Bosilovich et al. (2015), Molod et al. (2015), and Takacs et al. (2016). Some of the changes between MERRA and MERRA-2 that may affect UTLS dynamical fields are:
– New observation types have been added in MERRA-2, including hyperspectral infrared data from IASI (Infrared Atmospheric Sounding Interferometer) and CrIS (Cross-track Infrared Sounder), GPS-RO (Global Positioning System-Radio Occultation) bending angles, and polar wind observations from AVHRR (Advanced Very High Resolution Radiometer).
– MERRA-2 treats conventional temperature data differently, including changes in their error statistics and usage of adaptive bias correction for aircraft temperature data.
– Changes were made to the general circulation model, most notably a different horizontal grid and an improved convective parameterization scheme."

*p6, l7-11: I was wondering if the estimation of tropopause heights does involve any kind of higher-order interpolation (e.g. cubic spline) of the coarse-grid temperature profiles from the reanalyses on a fine vertical grid?*

No higher order interpolation is used because it may exaggerate extrema in regions of strong temperature gradients (such as often occur near the tropopause). A linear interpolation between the two levels on either side of the threshold is used to locate the tropopause between adjacent levels; we have added a sentence to this effect in the text (which has also been modified to address a comment from Dr. Añel).

*p6, l14: Why doesn't it make sense to construct means of frequency distributions from multiple reanalyses?*

To compare fields between multiple reanalyses, a "reanalysis ensemble mean" is often used, wherein the fields from each reanalysis are summed and the result divided by the number of reanalyses. What we meant to convey here is that using this procedure for frequency distributions does not produce a field that is useful for comparisons, since a mean constructed that way is no longer a frequency distribution, does not "weight" each reanalysis equivalently, and differences from that field would be problematic to interpret. Since we need a "reference" to take differences, we have chosen to use MERRA-2, the most recent of the reanalyses studied here. In the revised text, we state: "A reference distribution is needed to evaluate differences between the frequency distributions. However, taking a mean of the frequency distributions from the five reanalyses would result in a field that is problematic to interpret since it no longer represents a frequency distribution, and the reanalyses would not be equally weighted. Therefore, we have chosen to compare the other reanalyses to MERRA-2…."

*p6, l17-21 (and other places): It might be more clear if arithmetic differences of percentages would be referred to as "percentage points (pp)" rather than using "%" as the unit for these differences.*

This is a very good idea, and we have implemented it in the revised text. The discussion of arithmetic differences of percentages has been modified as follows:
"Because the frequency distributions are expressed as a percent (representing the fraction of the time there is a jet core, multiple tropopause, or subvortex jet in the bin, as discussed below in relation to normalization), the arithmetic differences (i.e., $Freq\_r1 - Freq\_r2$, where r1 and r2 are two reanalyses) between two frequency distributions that are shown in the figures are expressed as "percentage points" (pp); this should not be confused with the approximate percentage values for relative differences (e.g., $(Freq\ r1 - Freq\ r2)/0.5(Freq\ r1 + Freq\ r2) \times 100$) mentioned in the text...."

Also, the units of % for frequency distributions and pp for arithmetic differences of frequency distributions have been added to all of the figures. In the Figure 1 caption, we now state "In this

and all following figures, frequency distributions are expressed in percent (%) and arithmetic differences of frequency distributions in percentage points (pp)."

The statement in Section 3.2 regarding this has been modified as follows: "(Recall that, as described in Section 2.3, since frequency is expressed as a percent, the arithmetic differences between MERRA-2 and other reanalysis frequency distributions are expressed as percentage points (pp); the relative (percent) differences noted here are obtained by dividing the pp value in the difference plot by the percent value in the MERRA-2 frequency distribution plot.)"

*p8, l9-16: Can you provide a physical explanation of the northward shift of the jets between the MERRA-2 ASM and ANA fields?*

We have added the following statement to the text:
"Because of the IAU procedure used (see, e.g., Bloom et al. 1996, Fujiwara et al. 2016), the differences between ASM and ANA are to first order half of the analysis increment, with ASM being closer to the model results for a short forecast, and ANA (albeit less balanced) being closer to the observations. The ASM-ANA differences thus largely reflect small biases between the model and observations that develop over a short forecast period.  These might be expected to be qualitatively similar to the biases between the free-running model and the reanalysis. Molod et al. (2012) noted zonal mean wind biases between MERRA and a free-running GCM suggesting differences in both strength and position of the subtropical jet, as well biases in the eddy geopotential height fields that suggest regional variations in wind biases.  Biases of this sort persist between MERRA-2 and corresponding free-running models (Clara Orbe, personal communication) that appear broadly consistent with the shift of the jets seen here."

*p9, l12-21: Here I was also wondering if you could possibly provide a more detailed physical reasoning for the differences, the jet shifts in particular?*

A detailed analysis of the differences each observational input makes in the assimilated fields is beyond the scope of this paper.  However, the satellite radiance inputs play a large role in determining the temperature profiles, and without them the SH high latitude fields are poorly constrained by data. They directly and substantially affect the multiple tropopause distributions, and are also expected, via thermal wind balance, to play a large role in constraining the winds in the UTLS.  We have added a note to this effect in the text (at the end of the paragraph on the JRA-55 / JRA-55C comparisons): "Because the SH middle to high latitude fields are poorly constrained by conventional data, the assimilated satellite radiances are critical to constraining the temperature profiles here and, via thermal wind balance, are expected to be an important constraint for the wind fields as well.  Thus poor agreement in multiple tropopause distributions in SH middle to high latitudes, as well as larger differences in the jet distributions than in other regions, is consistent with expectations."

*Technical Corrections*

*p6, l14: doesn't -> does not*

The wording has already been changed per a previous comment.

*p10, l18-19: "in the both the" -> "in both the"*

Corrected.

*p12, l9-11: A verb seems to be missing in this sentence.*

The verb "show" has been added.

*p18, l1: The acronym "JETPAC" was not introduced before.*

We have modified the first sentence of section 2.2 to correct this omission, it now reads "The JETPAC (JEt and Tropopause Products for Analysis and Characterization) package described by Manney et al. (2011, 2014) is used here to characterize the UTLS jets and the tropopauses."

---

## Author Comment (AC3) · 4 Aug 2017

*This paper provides a comprehensive study of the dynamics of the UTLS regions as encapsulated in the jets and multiple tropopauses occurring in this region. The paper provides useful information to the atmospheric sciences community. The discussion of the performance of the reanalyses further provides information on their ability to represent the dynamics of the UTLS region. I recommend acceptance for ACP once the authors address the specific comments below.*

We thank the referee for their helpful comments.  The referee's comments are shown in blue italics; our responses are shown in black.  (Note that in addition to addressing the referees' comments, the figures have been re-done to use the recently updated "standard" S-RIP colors for each reanalysis.)

*Specific comments*

*P. 5 L. 23 and elsewhere: Is the standard convention ms-1 or m/s? Same for similar ratios C1(and see caption for Fig. 13).*

We have changed all occurrences to ms$^{-1}$, and, similarly, K/km to Kkm$^{-1}$.

*P. 6 L. 9-11: Perhaps rephrase, as the meaning of this sentence is not fully clear to this reviewer.*

This text has been rephrased to describe the definition and methods used here more explicitly, and text added to address the comments of Dr. Añel.  The revised text reads:
"If dT/dz drops below -2 Kkm^{-1} above the primary thermal tropopause, then the next level above that where the WMO criterion is fulfilled is identified as a multiple tropopause (Randel et al. 2007, Manney et al. 2011, Manney et al. 2014); this definition follows that of Randel et al. (2007), who showed that requiring dT/dz to drop only below -2Kkm^{-1} above the primary tropopause for the relatively coarse resolution reanalyses (rather than -3 Kkm^{-1} as is typically used for high-resolution temperature profiles) resulted in multiple tropopause distributions more comparable to those from high resolution measurements.  Linear interpolation is used to locate the tropopause between two adjacent vertical gridpoints.  Note that ``multiple tropopause" is used here to denote any profile with more than one tropopause.  As quantified by Schwartz et al. (2015), a very small fraction of the profiles have more than two tropopauses, and using only double tropopause versus all multiple tropopause profiles makes no significant difference in our results."

*L. 18: As a percent of what?*

We have added a parenthetical remark explaining this, thus: "...expressed as a percent (representing the fraction of the time there is a jet core, multiple tropopause, or subvortex jet in the bin, as discussed below in relation to normalization)". The immediately following wording has also been revised / clarified in response to a comment from referee #2.

*P. 8 L. 29: Why is this unsurprising?*

The much coarser vertical grid spacing (about 2 km for pressure coordinate versus less than 1 km for model level fields) leads to effective vertical smoothing of the fields, so that the magnitude of gradients and extrema can be underestimated. This is particularly likely to affect the representation of threshold phenomena using criteria based on vertical structure. We have changed the sentence in question to read "Because the much coarser vertical grid spacing can lead to underestimation of gradients and extrema, it is also unsurprising that a vertical spacing near 2 km in the UTLS…"

*P. 10 L. 18: Omit superfluous "the".*

This has been corrected.

*P. 14 Sect. 4, Summary and conclusions: This is an editorial decision, but I would suggest the authors consider splitting this section into two, a summary and a conclusion sections with the latter being succinct.*

We feel that the Summary and Conclusions section is already concise and focused, and do not see a natural way to divide into two sections that would provide further information or further clarify the results. We have therefore chosen to keep it as one section.

*P. 23 Fig. 1: Not clear to this reviewer which contour line corresponds to each percentage. I suggest the authors make this clearer in the figure, maybe by attaching the percentage to the contour (same for other figures, e.g., Fig. 2). I also suggest the authors consider providing information in the caption what the colours indicate, e.g., red/blue indicate positive/negative differences. Same for other figures.*

It would be difficult to add legible numbers to the contours on the figures, and would make the figures more "cluttered". However, we have modified the Fig. 1 caption to read: "...Overlaid contours show frequency values from each reanalysis of 10, 20, and 30% for upper tropospheric and subvortex jets, and 30, 45, and 60\% for multiple tropopauses; the smallest value is always the largest or ``outermost'' contour. In the difference plots, blues/oranges indicate negative/positive differences." We have added similar text to each caption where overlaid contours are described and difference plots are shown.

*P. 28 Fig. 6: The authors should clarify in the caption the differences between Figs. 5 and 6.*

To clarify this, we have changed the start of the Fig. 5 caption to:

"Seasonal maps of frequency distributions during JJA in 1980 through 2012 of JRA-55 (left) fields and the difference between JRA-55 and JRA-55C fields (right)."

And that of the Fig. 6 caption to:

"Seasonal latitude/altitude cross-sections of frequency distributions for JJA in 1980 through 2012 of JRA-55 fields (left) and the difference between JRA-55 and JRA-55C fields (right)."